# DGQ: Distribution-Aware Group Quantization for Text-to-Image Diffusion Models

**Hyogon Ryu    NaHyeon Park    Hyunjung Shim**[*]
Korea Advanced Institute of Science and Technology (KAIST)
{hyogon.ryu, julia19, kateshim}@kaist.ac.kr

## ABSTRACT

Despite the widespread use of text-to-image diffusion models across various tasks, their computational and memory demands limit practical applications. To mitigate this issue, quantization of diffusion models has been explored. It reduces memory usage and computational costs by compressing weights and activations into lower-bit formats. However, existing methods often struggle to preserve both image quality and text-image alignment, particularly in lower-bit($< 8$bits) quantization. In this paper, we analyze the challenges associated with quantizing text-to-image diffusion models from a distributional perspective. Our analysis reveals that activation outliers play a crucial role in determining image quality. Additionally, we identify distinctive patterns in cross-attention scores, which significantly affects text-image alignment. To address these challenges, we propose Distribution-aware Group Quantization (DGQ), a method that identifies and adaptively handles pixel-wise and channel-wise outliers to preserve image quality. Furthermore, DGQ applies prompt-specific logarithmic quantization scales to maintain text-image alignment. Our method demonstrates remarkable performance on datasets such as MS-COCO and PartiPrompts. We are the first to successfully achieve low-bit quantization of text-to-image diffusion models without requiring additional fine-tuning of weight quantization parameters. Code is available at https://github.com/ugonfor/DGQ.

## 1 INTRODUCTION

Diffusion models (Sohl-Dickstein et al., 2015) have recently become a key component of modern text-to-image models (Rombach et al., 2022; Ramesh et al., 2022; Li et al., 2023b; Podell et al., 2024), enabling the generation of high-quality images from natural language prompts. However, they often require a high computational workload, driven by iterative computations and significant memory costs. (Figure 1). This has limited their practicality in real-world applications, particularly on resource-constrained devices or in real-time generation (Kim et al., 2023; Ulhaq et al., 2022).

To reduce excessive computing resource usage, model quantization has gained significant attention. It involves compressing weights and activations from floating-point formats to lower-bit representations, thereby reducing both memory usage and computational requirements. Numerous approaches (Shang et al., 2023; Li et al., 2023a; He et al., 2023; Huang et al., 2024; So et al., 2024; He et al., 2024) have been proposed to quantize diffusion models while minimizing image quality degradation. However, these methods often fail to maintain accurate text-image alignment, which is crucial for text-to-image models (Figure 2). While reducing the degradation in text-image

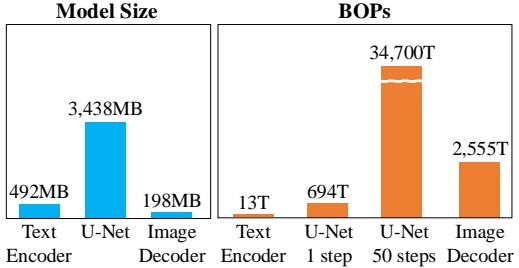

Figure 1: **Memory requirements and computational cost of Stable diffusion v1.4.**

---

[*]Hyunjung Shim is a corresponding author.

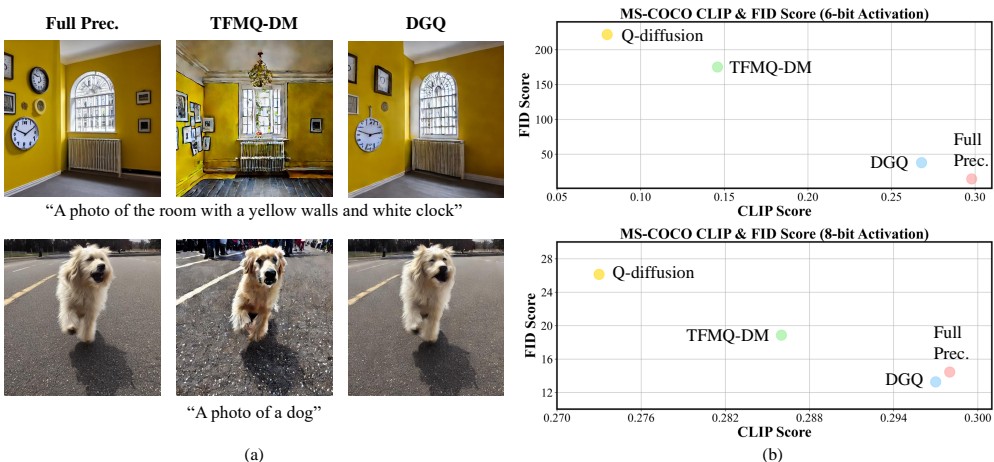

Figure 2: **The impact of DGQ.** (a) Two types of performance degradation in text-to-image diffusion model quantization. DGQ preserves both text-image alignment (as shown above) and image quality (as shown below) significantly better than TFMQ-DM. Each model is quantized to the 8-bits setting (both weight and activation). (b) Performance comparison with other methods.

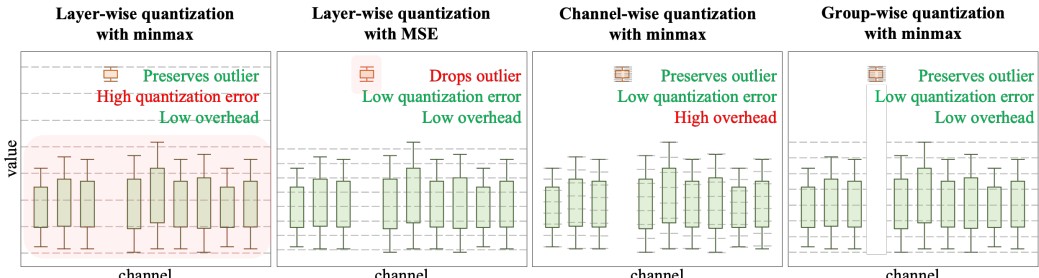

Figure 3: **Comparison of quantization strategies.** We show layer-wise, channel-wise and group-wise quantization methods. Minmax and MSE (mean-squared error) are the most common strategies for calibrating the quantization scale, but both approaches struggle to effectively quantize the activation. The gray dotted lines represent the quantized values. Unlike layer-wise quantization, in channel-wise quantization, the quantized values are adapted to each channel. In group-wise quantization, the quantized values are adapted to groups, such as outliers or other channels. More detailed information about quantization granularity can be found in Appendix E

alignment after quantization has been less explored, it is essential considering the application scenarios of text-to-image models. Most recently, Mixdq (Zhao et al., 2024) and PCR (Tang et al., 2023) have tried to quantize text-to-image diffusion models. These methods evaluated the sensitivity of each layer (Zhao et al., 2024) or timesteps (Tang et al., 2023) based on both image quality and text-image alignments, and allocate higher bit precision to more sensitive components. Both approaches, however, rely on mixed precision which presents challenges for hardware implementation. Additionally, they didn't focus on a lower-bit(below 8-bits) setting.

This paper investigates the quantization of text-to-image diffusion models, enabling hardware-friendly lower-bit precision quantization. While existing methods primarily focus on maintaining high image quality, our approach aims to preserve both high image quality and text-image alignment after quantization. We analyze the distribution of activations and identify key characteristics necessary for maintaining model performance during the quantization process. Firstly, we observe the presence of *outliers* in activations (**C1**), and recognize their critical role in preserving image quality. We also find that existing method using layer-wise quantization fail to retain these outliers. As a result, their generation performance is largely degraded even though quantization errors are successfully minimized (Figure 3).

We also observed that the attention scores in the cross-attention layers form a distinctive distribution (**C2**). As noted in many prior quantization studies on ViT (Lin et al., 2022; Li et al., 2023c), attention scores typically exhibit a simple log-normal distribution. However, our analysis revealed that,

unlike self-attention, cross-attention exhibits two distinct peaks each corresponding to `<start>` token and the remaining tokens, respectively. Existing methods disregard this distribution and employ a uniform quantizer, resulting in inadequate quantization of smaller values on a logarithmic scale. Moreover, due to the presence of the `<start>` token, the quantization error for the remaining tokens becomes significant, leading to degradation of text-image alignment. However, simply removing the `<start>` token significantly degrades image fidelity. Thus, we treat the `<start>` token separately, enabling us to achieve a high level of text-image alignment without compromising image quality.

Based on these findings, we propose Distribution-aware Group Quantization (DGQ) to address two key challenges in diffusion model quantization: (C1) outliers in activations and (C2) distinctive patterns in cross-attention, each having critical impact on image quality and text-image alignment. For **C1**, we introduce the outlier-preserving group quantization, which categorizes outliers into pixelwise and channel-wise outliers. Then, for each outlier type, we form a group and apply customized quantization with group-wise scale parameters. To address **C2**, we apply logarithmic quantization to the attention scores, except for `<start>` token, and use different quantization scales based on the input prompt. The attention score corresponding to `<start>` token is handled separately and maintained in full precision.

We tested our method on various datasets, including MS-COCO (Lin et al., 2014) and PartiPrompts (Yu et al., 2022), and confirmed its superior performance in generating high-quality and text-aligned images. Our method achieved a reduction of 1.29 in FID score compared to full precision and an almost identical CLIP score (a decrease of only 0.001) on MS-COCO dataset, while saving 93.7% in bit operations (from 694 TBOPs to 43.4 TBOPs).

To the best of our knowledge, we are the first to achieve low-bit quantization ($< 8$-bit) on text-to-image diffusion models (e.g., Stable Diffusion (Rombach et al., 2022)) without additional fine-tuning of weight quantization parameters.

In summary, our contributions are as follows:

- We identify that text-to-image diffusion models exhibit unique patterns in activations and cross-attention scores, which lead to performance degradation in existing quantization methods.
- We propose Distribution-aware Group Quantization (DGQ). It consists of outlier-preserving group quantization for activations and a customized quantizer for attention scores.
- Our outlier-preserving group quantization significantly enhances image quality after quantization, particularly in lower-bit settings. Meanwhile, customized quantizer for attentions facilitates high text-image fidelity.
- Extensive experiments demonstrate our method outperforms existing approaches. On the MS-COCO dataset, we achieve an FID score of 13.15, which is even lower than the score for full precision. Furthermore, we are the first to achieve lower-bit quantization (under 8-bit) in text-to-image diffusion models without any additional fine-tuning.

## 2 RELATED WORK

Diffusion models can successfully generate high-quality images through an iterative denoising process. Combined with pre-trained language models, diffusion models have shown outstanding performance in text-to-image generations. The release of large diffusion models such as Imagen (Saharia et al., 2022), Midjourney[1], DALL-E2 (Ramesh et al., 2022), GLIDE (Nichol et al., 2021), Stable Diffusion (Rombach et al., 2022), and FLUX[2] has further accelerated advancements in the field of generative AI. However, the high memory and computational costs of these large diffusion models present challenges for practical use.

Model quantization is a technique to reduce model size and improve inference speed by lowering the bit precision of the model's weights and activations. There are two primary approaches to model quantization: post-training quantization (PTQ) (Lin et al., 2024; Li et al., 2023a; Huang et al., 2024; Tang et al., 2023; Lin et al., 2022; He et al., 2024; Shang et al., 2023; Nagel et al., 2020; Wei et al.,

---

[1]https://midjourney.com/
[2]https://github.com/black-forest-labs/flux

2022; Li et al., 2021) or quantization-aware training (QAT) (Bondarenko et al., 2021; Esser et al., 2019; Jung et al., 2019). PTQ applies the quantization process after the model has been fully trained, requiring no additional training. In contrast, QAT incorporates the quantization process during training. It allows the model to adjust and maintain performance at lower precision. However, since QAT requires additional training time and resources compared to PTQ, PTQ is often more suitable for quantizing large foundation models. Recent works have a quantization-aware fine-tuning (He et al., 2023; Wang et al., 2024; Ryu et al., 2024; Kim et al., 2024; Dettmers et al., 2024) approach that slightly modifies QAT from-scratch and performs fine-tuning after PTQ. However, this ultimately requires additional computational cost. Our method has the advantage of not requiring such fine-tuning.

There have been studies on the quantization of diffusion models. These studies propose methods for quantization that take into account the timestep of diffusion models. Specifically, Q-Diffusion (Li et al., 2023a) constructs a calibration dataset by considering activation diversity across timesteps, while TFMQ-DM (Huang et al., 2024) employs a differently structured reconstruction block to better preserve temporal features. However, these studies focus solely on timestep-related characteristics without considering the text-condition. More recently, MixDQ (Zhao et al., 2024) measured layer-wise sensitivity with respect to the text condition and quantized through mixed precision. PCR (Tang et al., 2023) introduced a dynamic bit-precision mechanism depending on the timestep. However, both methods rely on mixed precision, making practical implementation challenging. A similar approach to ours, QuEST (Wang et al., 2024), highlights the importance of activation outliers and varies the quantization of weights. Different from these studies, our approach focuses on activation quantization, and both methods can be applied simultaneously.

## 3 METHOD

In this section, we provide an overview of hardware-friendly quantization techniques and analyze the challenges of applying existing methods to text-to-image diffusion models. To address these challenges, we introduce our approach, Distribution-aware Group Quantization (DGQ). DGQ is specifically designed to preserve the unique characteristics of activations and cross-attention scores. As a result, it maintains high image quality even at lower-bit settings and significantly improves text-to-image alignment.

### 3.1 PRELIMINARY: HARDWARE-FRIENDLY QUANTIZATION

We briefly introduce two hardware-efficient quantization methods: linear (uniform) quantization and logarithmic (log) quantization.

**Linear Quantization.** Linear quantization maps floating-point values to discrete integer levels uniformly across the value range. The quantization and de-quantization processes are defined as:

$$x_q = \text{clamp}\left(\left\lfloor \frac{x}{s} \right\rceil + z, 0, 2^b - 1\right), \quad x_{dq} = s \cdot (x_q - z) \approx x. \quad (1)$$

$x$, $x_q$, $x_{dq}$ are the floating-point input, quantized input, de-quantized input, respectively. $s$, $z$ and $b$ are the quantization parameter(scale factor, zero-point, bit-width). This method is popular due to its simplicity and compatibility with standard hardware operations.

**Logarithmic quantization.** Log quantization utilizes a logarithmic scale to handle values with a wide dynamic range, particularly effective for exponential distribution. The quantization and de-quantization processes are defined as:

$$x_q = \text{clamp}\left(\left\lfloor -\log_2\left(\frac{x}{s}\right) \right\rceil, 0, 2^b - 1\right), \quad x_{dq} = s \cdot 2^{-x_q} \approx x. \quad (2)$$

This method benefits from efficient hardware implementation using bit-shifting operations.

### 3.2 ANALYZING CHARACTERISTICS OF TEXT-TO-IMAGE DIFFUSION MODELS

To effectively quantize text-to-image models, we focus on analyzing them, particularly from a distributional perspective. Specifically, we examine the activations and attention scores, which we expect

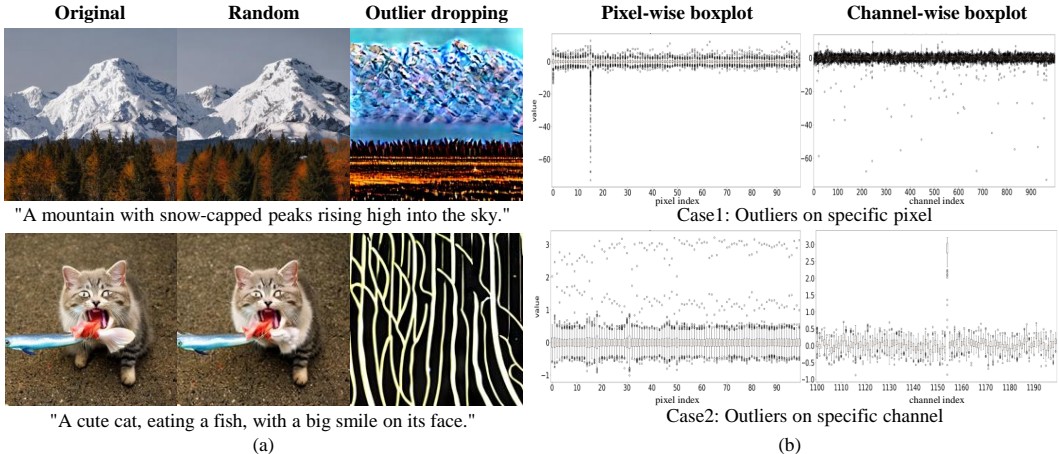

Figure 4: **Characteristics of activation outliers.** (a) Comparison of dropping random values and dropping outlier values. (b) Two types of outliers are identified. These outliers often appear in specific channels or at specific pixels. We provide full activation matrix visualization in Appendix H

to significantly impact on image quality and text-to-image alignment, repetively. Based on our investigation, we identified several key characteristics.

**Activation outliers play a crucial role in image quality.** We analyzed the importance of individual activation by examining the activation distribution of the diffusion model. Our analysis revealed that outliers (i.e., small fractions of activations with either very high or very low values) play a critical role in image generation. In Figure 4(a), we create three different images using the same text prompt with different activation manipulation. The first image represents the original generation (without any activations dropped), while the second and third images show examples where a single activation from each layer was set to zero. As illustrated in the figure, dropping random activations (that are not outliers) had minimal impact on the output image. However, dropping outlier activations resulted in images with drastically different shapes and noticeably lower quality (Table 1). This indicates that certain activations, specifically the outliers, are crucial for image generation.

Table 1: **Comparison of dropping values.** We used the original samples as reference images and evaluated them on a randomly sampled set of 1,000 MS-COCO prompts.

| Methods | PSNR↑ | LPIPS↓ |
|---|---|---|
| random drop | 17.81 | 0.295 |
| outlier drop | 9.34 | 0.773 |

Overall, our findings reveal that outliers significantly impact model performance, consistent with observations made in studies on large language models (Lin et al., 2024) and Vision Transformers (Darcet et al., 2023). Unlike these studies, our work primarily focuses on the effects of activations, particularly the role of activation outliers.

**Outliers appear on a few specific channels or pixels.** Then, where do outliers occur? We further trace the occurrence of outliers along various spatial dimensions. As shown in Figure 4(b), we confirmed that the outliers tend to appear in specific channels or pixels, rather than being evenly distributed. Furthermore, the locations of these outliers vary across different layers. This pattern persists even when the seeds and prompts are changed, suggesting that it is a distinctive characteristic of text-to-image diffusion models. We speculate that this results from specific architectural choices and long pretraining without activation regularization (Bondarenko et al., 2021).

**Cross-attention score corresponding to `<start>` token make a distinct peak.** Since attention scores are computed using the Softmax function, they typically follow a logarithmic distribution:

$$\text{Attention Score}(\mathbf{Q}, \mathbf{K})_{ij} = \text{Softmax}(\frac{s_{ij}}{\sqrt{d}}) = \frac{\exp\left(\frac{s_{ij}}{\sqrt{d}}\right)}{\sum_{j'} \exp\left(\frac{s_{ij'}}{\sqrt{d}}\right)}, \quad \text{where } s_{ij} = \mathbf{Q}_i \cdot \mathbf{K}_j^T. \quad (3)$$

$\mathbf{Q} \in \mathcal{R}^{n_q \times d}, \mathbf{K} \in \mathcal{R}^{n_k \times d}, i = 1, 2, ..., n_q$, and $j = 1, 2, ..., n_k$. $n_q$ and $n_k$ denotes number of tokens, and $d$ is the feature size. Since $q_i$ and $k_j$ are normally distributed in self-attention, $\frac{s_{ij}}{\sqrt{d}}$ is also normally distributed, and the exponentials $\exp\left(\frac{s_{ij}}{\sqrt{d_k}}\right)$ follow a log-normal distribution. Consequently,

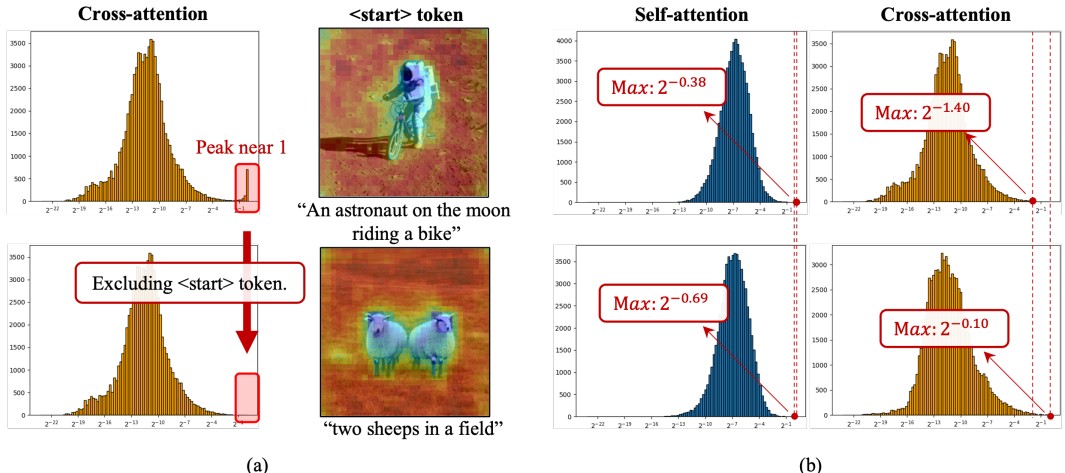

Figure 5: **Characteristics of cross-attention scores.** (a) The `<start>` token causes a peak near 1.0(Left). Background pixels tend to have high attention scores for the `<start>` token (Right). (b) Unlike self-attention, the maximum values of cross-attention scores change more dynamically.

in log scale, self-attention scores are approximated by a normal distribution. However, when examining the distribution of cross-attention scores, distinct patterns emerge, where they clearly differ from that of self-attention scores. As shown in Figure 5(a), the first notable difference is the presence of a peak near the `<start>` (also referred to as `<bos>`) token. To find the cause of this pattern, we analyze the attention score of the `<start>` token. We find that the background pixels tend to have high attention scores (almost close to 1.0) for the `<start>` token. Moreover, we empirically confirmed the role of these high attention scores of the `<start>` token. Adjusting attention scores to various levels (e.g., dropping to zero and clamping to the second highest token's attention score) affects overall image details. (See Appendix A).

**The distribution of cross-attention scores is highly dependent on the input prompts.** Additionally, we analyzed the attention scores except for `<start>` token and observed that the distribution of the cross-attention scores are also different from those of self-attention. An image input has fixed pixel size and locality, resulting in consistently widespread attention scores and a similar distribution range across the input prompts. In contrast, the number of text tokens relevant to specific pixels varies, causing cross-attention scores to either concentrate or disperse. As shown in Figure 5(b), this variation in distribution depends on the input prompts (Statistical results are in Appendix G). The high variation makes it difficult for a static quantizer to preserve large values, resulting in information loss in the content at the pixel level, which is important for text-image alignment.

### 3.3 DISTRIBUTION-AWARE GROUP QUANTIZATION

Based on our findings on activation and cross-attention scores, we develop a novel quantization method tailored to text-to-image diffusion models. Specifically, we suggest (1) outlier-preserving group quantization for handling outliers in activations and (2) attention-aware quantization for addressing patterns in cross attention.

**Outlier-preserving group quantization.** To maintain high image quality during quantization, it is essential to preserve outliers while minimizing quantization error. However, most existing approaches fail to meet both of these criteria at the same time. As shown in Figure 3, when outliers are preserved, the overall quantization error becomes excessively large. Conversely, if the quantization scale is optimized to minimize mean-squared error, outliers are often ignored. Although channel-wise quantization can mitigate this issue, it introduces significant computational overhead and is ineffective at handling pixel-level outliers.

We propose outlier-preserving group quantization, that effectively reduces computational overhead while maintaining image quality. Our approach identifies the most efficient dimension for applying group quantization in each layer. After considering the activation range for vectors along the selected dimension, we group them accordingly and create a customized quantizer for each group. Specifi-

cally, for each layer, we identify the outlier type by examining the activation range across channels or pixels. Since outliers appear in specific channels or pixels, the difference in activation range in the dimensions, where outliers occur, is larger than in other dimensions. We determine the optimal dimension $d \in \{channel, pixel\}$ and apply quantization in a way that preserves critical outlier information. To achieve this, we define a metric $D_d$ that measures the variability of activation values in each dimension:

$$D_d = \left( \max_i a_{i,d}^{\max} - \min_i a_{i,d}^{\max} \right) + \left( \max_i a_{i,d}^{\min} - \min_i a_{i,d}^{\min} \right), \tag{4}$$

where $a_{i,d}^{\max}$ and $a_{i,d}^{\min}$ represent the maximum and minimum values of the $i$-th vector in dimension $d$, respectively. $i$ is the index of the $d$-th dimension. This metric indicates the magnitude of the differences in activation value ranges across the channel and pixel dimensions

Then, the dimension $d^*$, where $D_d$ is large, is selected as the dimension to be used for grouping:

$$d^* = \arg \max_d D_d. \tag{5}$$

At the optimal dimension $d^*$, we divide the activation values into $K$ groups, based on their range using K-means clustering. For each group, quantization scale $s_k$ and zero-point $z_k$ are calculated as:

$$s_k = \frac{\max \mathbf{x} - \min \mathbf{x}}{2^b}, \quad z_k = \min \mathbf{x}, \tag{6}$$

where $k \in \{1, ..., K\}$. $\mathbf{x}$ represents the activation belonging to the $k$-th group, and $b$ denotes the number of quantization bits. Each group quantizes its values using the same quantizer. This approach adjusts the quantization scale according to the distribution range of the activation values within each group. This minimizes quantization errors and preserves outliers, effectively preserving image quality.

Finally, we consider the variance of activation that changes with the timestep of the diffusion model, we consider the same process for each timestep as described in previous studies (He et al., 2023; Huang et al., 2024). The overall quantization scale $S$ and zero-point $Z$ for each layer are as follows:

$$S = \{s_0, s_1, ..., s_{T-1}\}, \quad Z = \{z_0, z_1, ..., z_{T-1}\}. \tag{7}$$

Note that $T$ represents the total denoising step. For Stable Diffusion v1.4, when $T$ is set to 25 steps and 16 groups are used, the additional memory occupied by the quantization parameter is $25 \times 16 \times 3008(\text{byte}) = 2.29\text{MB}$. This is negligible($\simeq 0.1\%$) compared to the UNet's memory requirement of 3,438 MB.

**Attention-aware quantization.** Cross-attention plays a crucial role in aligning text and images, as it integrates text conditions into the image generation model (Zhao et al., 2024; Wang et al., 2024). As discussed in Section 3.2, the distribution of attention scores has several clearly different patterns from other activations. However, existing methods naively use a uniform quantizer for handling these attentions. They therefore fail to preserve this distribution and leading to text-image alignment degradation. To address this problem, first, we apply a logarithmic quantizer to both self and cross-attentions. In this way, we can preserve the small values in log scale while uniform quantizer cannot. Secondly, for cross-attention, we separate the forward path for the attention scores corresponding to the <start> token and the others. Then, we maintain the attention scores of <start> token and apply the quantizer to attention scores of the others. Since the Softmax operation is normally performed in full precision, no additional dequantization is needed, making it efficient to implement in hardware. Third, since the range of cross-attention scores varies depending on the input prompt, we employ dynamic quantization, which adjusts the quantization scale to the maximum value of the attention score excluding <start> token in inference-time. Our quantization process can be expressed as the multiplication between quantized attention score $\hat{\mathbf{A}} \in \mathcal{R}^{n_q \times n_k}$ and quantized value $\hat{\mathbf{V}} \in \mathcal{R}^{n_v \times d}$. That is,

$$\mathbf{A}_{[:,1:]}^q = \text{clamp} \left( \left\lfloor -\log_2 \left( \frac{\mathbf{A}_{[:,1:]}}{s} \right) \right\rceil, 0, 2^b - 1 \right), \quad \text{where } s = \max(\mathbf{A}_{[:,1:]}) \tag{8}$$

$$\hat{\mathbf{A}} = [\mathbf{A}_{[:,0]}, s \cdot 2^{-\mathbf{A}_{[:,1:]}^q}], \quad \hat{\mathbf{A}}\hat{\mathbf{V}} = \left[ \mathbf{A}_{[:,0]} \hat{\mathbf{V}}_{[0,:]}, s \cdot 2^{-\mathbf{A}_{[:,1:]}^q} \hat{\mathbf{V}}_{[1:,:]} \right], \tag{9}$$

where $\mathbf{A}$ is the full precision attention score. $n_q$, $n_k$, $n_v$ are the number of tokens of query, key, and value, respectively.

| Model | Method | Bits(W/A) | Model Size | BOPs | MS-COCO | | | PartiPrompts |
|---|---|---|---|---|---|---|---|---|
| | | | | | IS↑ | FID↓ | CLIP↑ | CLIP↑ |
| **SD v1.4** | **Full Precision** | 32/32 | 3,438MB | 823T | 36.52 | 14.44 | 0.298 | 0.293 |
| | **Q-Diff** | 8/8 | 871MB | 51.4T | 27.65 | 26.12 | 0.273 | 0.275 |
| | **TFMQ** | 8/8 | 871MB | 51.4T | 32.79 | 18.85 | 0.286 | 0.286 |
| | **DGQ** (#groups=8) | 8/8 | 871MB | 51.4T | **35.38** | 13.26 | **0.297** | **0.292** |
| | **DGQ** (#groups=16) | 8/8 | 871MB | 51.4T | 35.22 | **13.15** | **0.297** | **0.292** |
| | **Q-Diff** | 8/6 | 871MB | 38.6T | 4.12 | 221.76 | 0.080 | 0.120 |
| | **TFMQ** | 8/6 | 871MB | 38.6T | 6.57 | 175.16 | 0.146 | 0.178 |
| | **DGQ** (#groups=8) | 8/6 | 871MB | 38.6T | 22.65 | 37.76 | 0.268 | 0.277 |
| | **DGQ** (#groups=16) | 8/6 | 871MB | 38.6T | **24.77** | **31.36** | **0.273** | **0.279** |
| | **Q-Diff** | 4/8 | 436MB | 25.7T | 26.52 | 28.06 | 0.269 | 0.271 |
| | **TFMQ** | 4/8 | 436MB | 25.7T | 30.85 | 19.98 | 0.281 | 0.281 |
| | **DGQ** (#groups=8) | 4/8 | 436MB | 25.7T | **33.91** | **13.28** | **0.294** | **0.289** |
| | **DGQ** (#groups=16) | 4/8 | 436MB | 25.7T | 33.56 | 13.74 | **0.294** | 0.288 |
| | **Q-Diff** | 4/6 | 436MB | 19.3T | 3.37 | 242.75 | 0.072 | 0.108 |
| | **TFMQ** | 4/6 | 436MB | 19.3T | 5.24 | 229.64 | 0.127 | 0.155 |
| | **DGQ** (#groups=8) | 4/6 | 436MB | 19.3T | 20.14 | 51.94 | 0.257 | 0.272 |
| | **DGQ** (#groups=16) | 4/6 | 436MB | 19.3T | **22.17** | **43.66** | **0.263** | **0.274** |
| **SDXL Turbo** (4 steps) | **Full Precision** | 32/32 | 10,269MB | 6,927T | 35.97 | 21.25 | 0.308 | 0.309 |
| | **TFMQ** | 8/8 | 2,567MB | 433T | 12.24 | 111.69 | 0.067 | 0.069 |
| | **DGQ** (#groups=8) | 8/8 | 2,567MB | 433T | **34.79** | **22.46** | **0.299** | **0.294** |
| | **TFMQ** | 8/6 | 2,567MB | 325T | 4.27 | 163.02 | -0.002 | 0.025 |
| | **DGQ** (#groups=8) | 8/6 | 2,567MB | 325T | **28.56** | **34.31** | **0.251** | **0.223** |
| | **TFMQ** | 4/8 | 1,284MB | 216T | 13.00 | 109.56 | 0.068 | 0.069 |
| | **DGQ** (#groups=8) | 4/8 | 1,284MB | 216T | **28.33** | **29.22** | **0.289** | **0.291** |
| | **TFMQ** | 4/6 | 1,284MB | 162T | 1.99 | 270.45 | 0.022 | 0.049 |
| | **DGQ** (#groups=8) | 4/6 | 1,284MB | 162T | **22.93** | **45.00** | **0.245** | **0.226** |

Table 2: **Quantitative Comparison.** Results of different quantization methods on MS-COCO and PartiPrompts datasets.

## 4 Experiments

### 4.1 Implementation details

**Datasets, models and evaluation metrics.** The dataset used for calibration during quantization was generated using 64 captions from the MS-COCO Dataset (Lin et al., 2014). Similar to the approach taken in Tang et al. (2023), we evaluated prompt generalization performance using the PartiPrompts (Yu et al., 2022) dataset, which differs from the calibration dataset. For the text-to-image model, we used Stable Diffusion v1.4. We measured FID (Heusel et al., 2017) and IS (Salimans et al., 2016) scores to evaluate image quality, and the CLIP score to evaluate text-image alignment. For main results (Table 2), we compute the FID and IS using 30K samples. For the ablation study (Table 3), we use 10K samples. Additionally, to evaluate computational cost, we measured BOPs (BOPs $= \text{FLOPs} \cdot b_w \cdot b_a$), where $b_w$ and $b_a$ each stand for bits of weight and activation, respectively.

**Baseline and implementation details.** We use two state-of-the-art methods, Q-Diffusion (Li et al., 2023a) and TFMQ-DM (Huang et al., 2024), as baselines for comparison. To ensure a fair evaluation, we employ the diffusers [3] library for both the baselines and our method. Unless specified otherwise, we apply 25 inference steps for computational efficiency. It should be noted that Q-Diffusion and TFMQ-DM set the attention score's quantizer bits to 16 bits to avoid text-image alignment degradation. However, in our implementation, to ensure a fair comparison, we set all attention score's quantizer bits to match the activation bits.

**Weight quantization.** Since our method focuses on activation quantization, we evaluated its effectiveness by applying the same quantization methods to the weights of both the baseline and our method. Following previous studies Huang et al. (2024); Li et al. (2023a); Shang et al. (2023); Tang et al. (2023), we used BRECQ (Li et al., 2021) and Adaround (Nagel et al., 2020) for weight quantization. Block reconstruction were applied to both transformer and residual blocks. The calibration dataset used for reconstruction was the same as that used for activation quantization. We collect the intermediate output with 64 captions from the MS-COCO dataset.

---

[3]https://github.com/huggingface/diffusers

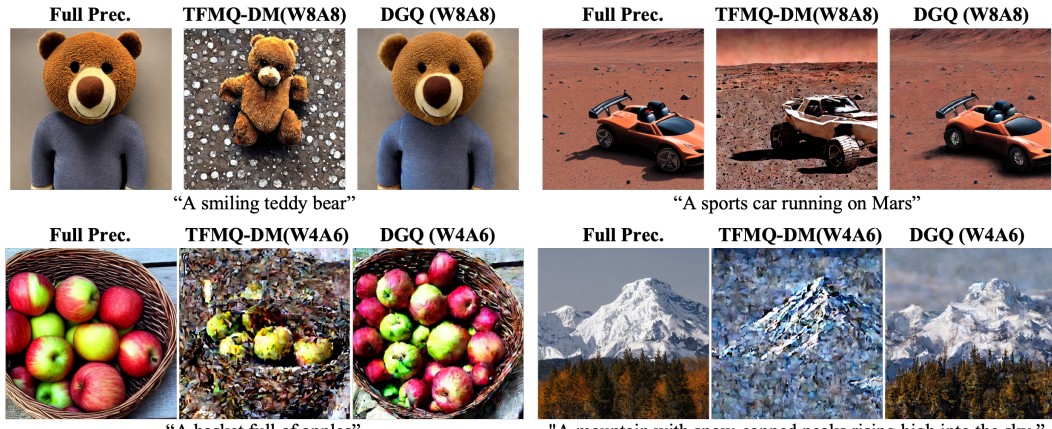

Figure 6: **Qualitative Comparison.** Images in the top row were generated with the W8A8 setting, and images in the bottom row were generated with the W4A6 setting. WXAY represents weights and activations with X and Y bits, respectively.

## 4.2 MAIN RESULTS

We conducted experiments on the MS-COCO and PartiPrompts datasets, with the results presented in Table 2. First, on the MS-COCO dataset, all results indicate that our DGQ significantly outperforms existing methods. Specifically, with 8-bit activations, DGQ consistently delivers the best performance, regardless of the weight bit width. The FID scores were 13.15 and 13.28 for the W8A8 and W4A8 settings, respectively, surpassing the performance of the full-precision model (14.44). For text-image alignment, the CLIP score experienced only minimal drops (less than 0.005).

For the 6-bit activation setting, baseline methods failed to generate images, whereas DGQ successfully produced images of acceptable quality. Although there was inevitable performance degradation compared to the full-precision model, our results showed significant improvement over the baseline methods. These improvements were evident in both the FID score (reducing from over 200 to 43.66 in the W4A6 setting) and the CLIP score (increasing from 0.155 to 0.274 in the W4A6 setting).

Our method outperformed all previous approaches when evaluated on the PartiPrompts dataset, which is designed to assess prompt generalization performance. Despite the PartiPrompts dataset includes different types of prompts compared to the captions in MS-COCO, we achieved a high CLIP score, suggesting our successful preservation of text-to-image alignment. The qualitative results can be seen in Figure 6.

## 4.3 ABLATION STUDY

We performed an ablation study to analyze the impact of each component of our proposed method. The experimental results are presented in Table 3.

**Effects of each component.** In this section, we analyze the effect of each component of DGQ. We examined the impact of Outlier-preserving Group Quantization and Attention-aware Quantization individually, and the results are shown in Table 3(a). For Outlier-preserving Group Quantization, a group size of 8 was used. For Attention-aware Quantization, we applied a Log Quantizer, separating the `<start>` token, and utilized dynamic quantization. Each component contributed to improvements in image quality and text-image alignment, with the performance boost from Outlier-preserving Group Quantization being particularly significant. The best performance was achieved when both techniques were applied together.

**Effects of grouping strategy.** We investigated the effects of group size and dimension selection in outlier-preserving group quantization. As shown in Table 3(b), the best image quality was achieved when dimension selection was applied and the group size was set to 2. Meanwhile, the best text-image alignment performance occurred when dimension selection was applied with a group size of 8. This suggests that increasing the group size does not always lead to better performance, and

| Bits(W/A) | Outlier | Attention | IS↑ | FID↓ | CLIP↑ |
|---|---|---|---|---|---|
| 8/8 | ✗ | ✗ | 31.34 | 18.83 | 0.286 |
| 8/8 | ✓ | ✗ | 32.76 | 17.00 | 0.296 |
| 8/8 | ✗ | ✓ | 31.18 | 18.24 | 0.287 |
| 8/8 | ✓ | ✓ | **33.61** | **14.40** | **0.297** |
| 8/6 | ✗ | ✗ | 6.45 | 176.44 | 0.144 |
| 8/6 | ✓ | ✗ | 18.20 | 50.04 | 0.250 |
| 8/6 | ✗ | ✓ | 7.17 | 207.46 | 0.127 |
| 8/6 | ✓ | ✓ | **21.88** | **40.15** | **0.267** |

(a) Effects of each component

| Dim. | # Groups | IS↑ | FID↓ | CLIP↑ |
|---|---|---|---|---|
| ✗ | 1 | 31.34 | 18.83 | 0.286 |
| ✗ | 2 | 32.45 | 16.74 | 0.290 |
| ✗ | 4 | 33.01 | 16.54 | 0.294 |
| ✗ | 8 | 32.61 | 16.79 | 0.294 |
| ✗ | 16 | 32.07 | 16.91 | 0.294 |
| ✓ | 2 | 32.94 | **16.24** | 0.292 |
| ✓ | 4 | **33.03** | 16.64 | **0.295** |
| ✓ | 8 | 32.64 | 16.99 | **0.295** |
| ✓ | 16 | 32.42 | 17.15 | **0.295** |

(b) Effects of grouping strategy

| Bits(W/A) | Quantizer | Seperate `<start>` | IS↑ | FID↓ | CLIP↑ |
|---|---|---|---|---|---|
| 8/8 | Linear | ✗ | 31.34 | 18.83 | 0.286 |
| 8/8 | Linear | ✓ | **31.44** | 18.78 | 0.285 |
| 8/8 | Log (w/ dynamic quant.) | ✗ | 30.91 | 18.37 | **0.287** |
| 8/8 | Log (w/ dynamic quant.) | ✓ | 31.18 | **18.24** | **0.287** |
| 8/8 | Log (w/o dynamic quant.) | ✗ | 24.51 | 31.12 | 0.265 |
| 8/8 | Log (w/o dynamic quant.) | ✓ | 24.60 | 28.79 | 0.271 |
| 8/6 | Linear | ✗ | 18.20 | 50.04 | 0.249 |
| 8/6 | Linear | ✓ | 17.64 | 49.98 | 0.249 |
| 8/6 | Log (w/ dynamic quant.) | ✗ | 21.36 | 41.28 | **0.268** |
| 8/6 | Log (w/ dynamic quant.) | ✓ | **21.88** | **40.15** | 0.267 |

(c) Effects of attention-aware quantization.

Table 3: **Ablation study.** In (c), Due to the low-quality image under the 8/6-bit setting, evaluation was not possible. Therefore, we applied outlier-preserving group quantization beforehand to analyze the impact of attention-aware quantization.

there exists an optimal group size. We interpret this phenomenon as being influenced by the 8-bit environment used in the experiments, where the quantization scale is already sufficiently small. As a result, increasing the group size further does not significantly enhance the precision of the activation representation. The drop in performance may be attributed to overfitting to the calibration dataset as the group size increases. In 8-bit settings, the best solution was to use 2 groups with dimension selection. However, we anticipated that a larger group size would be more effective in lower-bit settings. Therefore, we set the default group sizes to 8 and 16. As shown in Table 2, the results confirmed that in lower-bit settings, a group size of 16 outperformed a group size of 8 in both image quality and text-image alignment. More ablation study on 6-bit setting can be found in Appendix B

**Effects of attention-aware quantization.** Table 3(c) illustrates the impact of each component of attention-aware quantization. The best performance was observed when applying all components(Log quantizer, separating `<start>` token, and dynamic quantization). Separating the `<start>` token resulted in a slight reduction in the FID score. For Log quantizer without dynamic quantization, the quantization scale is determined using the *running-minmax* method. It calculates the min-max range through the exponential moving average of multiple batches (Krishnamoorthi, 2018). This method is commonly used to calibrate the log quantizer in existing vision transformer quantization approaches (Li et al., 2023c; Li & Gu, 2023), but we find that it is not suitable for the cross-attention in diffusion models. The log quantizer with dynamic quantization outperforms the linear quantizer; in particular, on a 6-bit activation setting, it outperforms the linear quantizer by a large margin.

## 5 CONCLUSION

In this work, we propose Distribution-aware Group Quantization (DGQ) for text-to-image diffusion models. We identify the crucial role of outliers in image quality and preserve them by grouping channels or pixels based on activation distribution. Furthermore, we uncover unique patterns in cross-attention scores and apply prompt-specific logarithmic quantization. DGQ outperforms existing methods and, for the first time, enables low-bit quantization of text-to-image diffusion models without additional fine-tuning. By reducing computational costs while preserving both image quality and text-image alignment, our approach broadens the deployment of diffusion models in real-world applications, including edge devices.

## ACKNOWLEDGEMENTS

We thank all CVML members for valuable feedbacks. This work was supported by Institute of Information & communications Technology Planning & Evaluation (IITP) grant funded by the Korea government(MSIT) (RS-2019-II190075 Artificial Intelligence Graduate School Program(KAIST), No.2021-0-02068 Artificial Intelligence Innovation Hub, No. RS-2024-00457882, National AI Research Lab Project), IITP grant funded by the Korea government(MSIT) and KEIT grant funded by the Korea government(MOTIE) (No. 2022-0-00680, 2022-0-01045), and Basic Science Research Program through the National Research Foundation of Korea (NRF) funded by the MSIP (NRF-2022R1A2C3011154, RS-2023-00219019).

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

## A  EFFECTS OF THE ATTENTION SCORE CORRESPONDING TO <START> TOKEN.

We analyze the effects of the attention score corresponding to <start> token. For that, we adjust the attention scores, and compare the sampled images. We change the attention score of <start> token in two ways, clamping and dropping. clamping sets the attention score to the maximum value of attention score corresponding to the other tokens(except <start> token), and dropping sets it to 0. Clamping is used to check whether this can be excluded when determining the quantization scale of the quantizer, and dropping is used to check whether this can be excluded altogether. We confirmed that the <start> token doesn't change the main contents of the images, but it affects on the details of the images. Therefore, in order to maintain full precision output as much as possible, <start> token should be preserved.

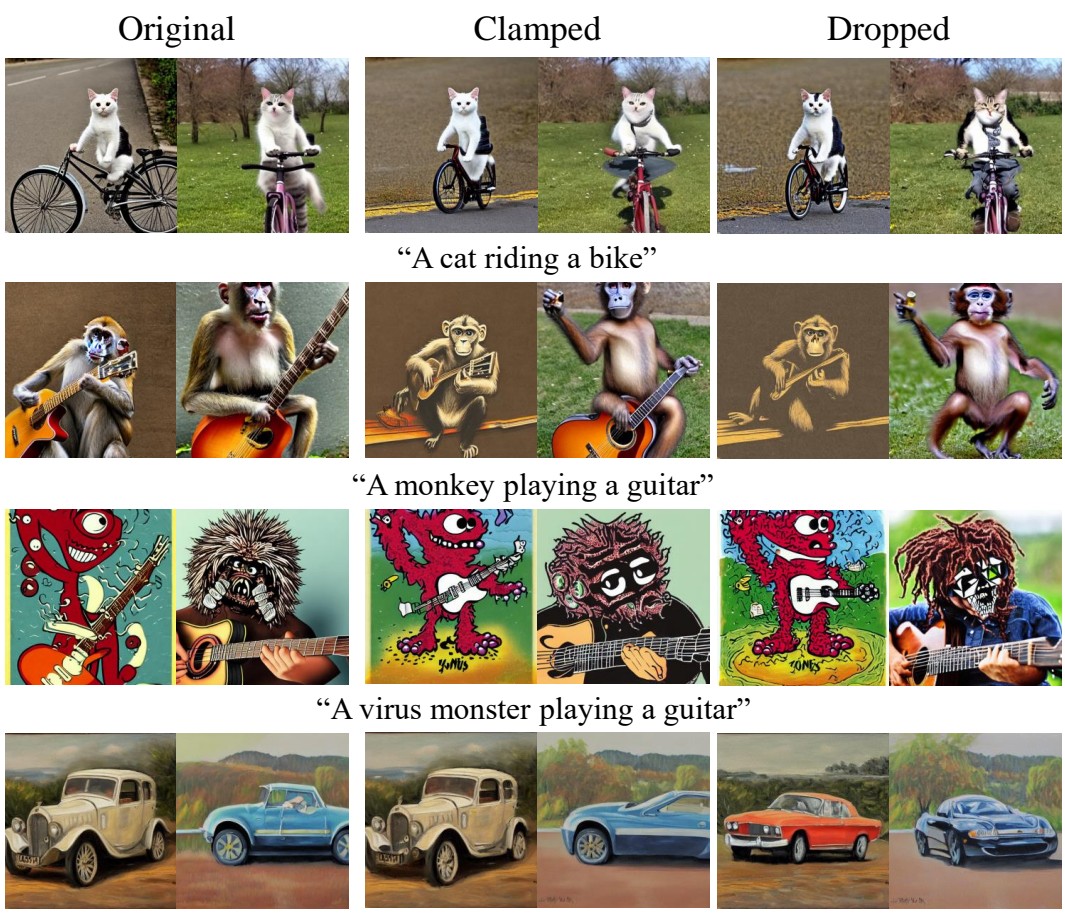

Figure A.1: **Analysis on <start> token.**

## B    FURTHER ABLATION STUDY ON GROUPING STRATEGY

We conduct an ablation study to assess the impact of different grouping strategies on a 6-bits setting. As shown in Table A.1, increasing the number of groups generally improves model performance. Applying dimension selection consistently yields better results, and, unlike the 8-bit setting, more groups consistently improve model performance.

Table A.1: **Effects of grouping strategy on 8/6 setting.**

| Dim. | # Groups | IS↑ | FID↓ | CLIP↑ |
|:---:|:---:|:---:|:---:|:---:|
| ✗ | 1 | 6.45 | 176.44 | 0.144 |
| ✗ | 2 | 8.44 | 136.14 | 0.190 |
| ✗ | 4 | 11.88 | 86.10 | 0.223 |
| ✗ | 8 | 14.06 | 70.35 | 0.232 |
| ✗ | 16 | 15.20 | 64.85 | 0.235 |
| ✓ | 2 | 9.69 | 117.11 | 0.201 |
| ✓ | 4 | 14.91 | 66.33 | 0.236 |
| ✓ | 8 | 17.70 | 52.11 | 0.243 |
| ✓ | 16 | **18.46** | **49.05** | **0.248** |

## C    EVALUATION ON VARIOUS METRICS

Considering the widespread usage of image quality assessment (IQA) models and human preference reward models, we evaluate our methods on the IQA model MANIQA (Yang et al., 2022) and the human preference model ImageReward (Xu et al., 2024). The evaluation is conducted on 30K samples from the MS-COCO dataset. As shown in Table A.2, in almost all cases, DGQ significantly outperforms the baseline. With 8-bit activation settings, TFMQ achieves slightly higher performance with MANIQA, but on the ImageReward model, DGQ achieves better results in all cases.

Table A.2: **Quantitative comparison.** MANIQA is the image quality assessment model and ImageReward is the human preference reward model.

| Method | Bits(W/A) | MANIQA↑ | ImageReward↑ |
|:---|:---:|:---:|:---:|
| **Full Precision** | 32/32 | 0.5525 | 0.1120 |
| **TFMQ** | 8/8 | **0.5359** | -0.0375 |
| **DGQ**(#groups=8) | 8/8 | 0.5340 | **0.0668** |
| **TFMQ** | 8/6 | 0.3795 | -1.8643 |
| **DGQ**(#groups=8) | 8/6 | **0.4187** | **-0.2724** |
| **TFMQ** | 4/8 | **0.5265** | -0.1222 |
| **DGQ**(#groups=8) | 4/8 | 0.5223 | **-0.0256** |
| **TFMQ** | 4/6 | 0.3834 | -2.0587 |
| **DGQ**(#groups=8) | 4/6 | **0.4282** | **-0.4630** |

## D    DISCUSSION

### D.1    LIMITATION AND FUTURE DIRECTIONS

We summarize the current limitation and potential future works.

**Combining with advanced weight quantization methods.** Since our methods are concentrate on the activation quantization, it would be able combined with other advanced weight quantization techniques such as EfficientDM, QuEST or the other quantization-aware training methods.

**More effective quantizer for attention scores.** For cross-attention score, our analysis reveals that the distribution range are deeply depends on the user input prompts. In DGQ, because of hardware-constraint, we adjust the quantization scale as the maximum value of remaining attention scores, but it would be more effective methods such as using lookup table or reparameterization.

## D.2 DIFFERENCE BETWEEN POST-TRAINING QUANTIZATION AND QUANTIZATION-AWARE TRAINING

Some other quantization methods (Zheng et al., 2024; Sui et al., 2024) achieve extremely low-bit quantization. However, they are based on Quantization-Aware Training (QAT), which is a completely different setting from ours (i.e., Post-Training Quantization). BitFusion (Sui et al., 2024) requires a huge dataset and significant computational cost to obtain a quantized model. In contrast, DGQ (Ours) is a model generated through PTQ(Post-Training Quantization) that does not require a dataset, requires only 64 prompts, and has a minimal computational cost. Specifically, according to the BitFusion paper, their model was trained for 50K iterations with a batch size of 1024 using an internal dataset, utilizing 32 NVIDIA A100 GPUs. On the other hand, DGQ used only 64 sample prompts during the activation quantization process and was completed in about 20 minutes on just one RTX A6000 (based on Stable Diffusion v1.4 with 25 steps).

Generally, models quantized through QAT have better performance compared to models quantized through PTQ. However, due to the need for a huge training dataset and high training costs, QAT is not practical. Therefore, recent quantization research for the large foundation models has been focused on PTQ(please refer to the survey paper (Zhu et al., 2023)). DGQ is the first method to achieve low-bit quantization of text-to-image diffusion models without any additional fine-tuning (i.e., PTQ).

# E QUANTIZATION GRANULARITY

Quantization in deep learning models involves reducing the precision of weights and activations to lower bit-width representations, thereby enhancing computational efficiency and reducing memory consumption. The granularity of quantization—that is, the level at which quantization parameters are applied—significantly impacts the trade-off between model accuracy and computational performance. The primary types of quantization granularity are layer-wise, group-wise, and channel-wise quantization.

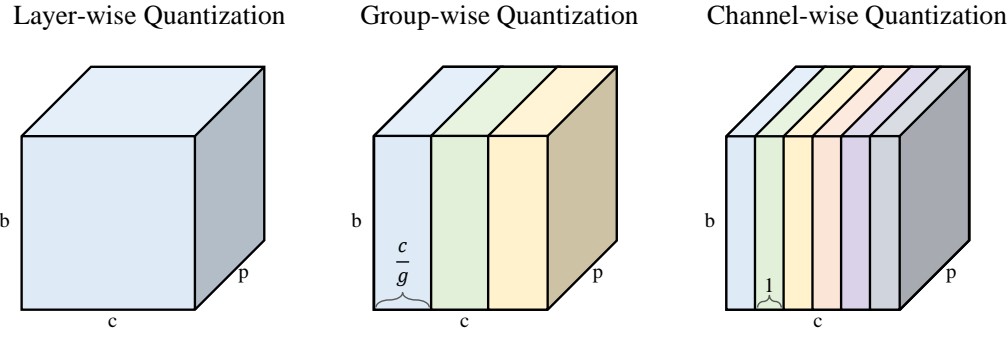

b: batch index, c: channel index, p: pixel index, g: group size

Figure A.2: **Illustration of Quantization Granularity.** Different quantizers are applied to each color.

As shown in Figure A.2, layer-wise quantization applies a single scale and zero-point to all weights or activations within an entire layer, simplifying implementation but potentially reducing accuracy due to its coarse approach. Group-wise quantization divides the weights or activations within a layer into multiple groups, assigning separate quantization parameters to each group. This method offers a balance between efficiency and precision, capturing more detail than layer-wise quantization without the full complexity of channel-wise quantization. Channel-wise quantization assigns individual

quantization parameters to each channel, providing the most precise representation of weight and activation distributions. While this fine-grained approach often yields higher model accuracy, it comes with increased computational and memory overhead due to the need to store and process multiple sets of quantization parameters.

## F IMPACT OF ATTENTION SCORE QUANTIZER BIT-WIDTH ON IMAGE-TEXT ALIGNMENT.

To analyze the effect of the attention score quantizer's bit-width, we adjust its bit-width while keeping the other layers at full precision. As shown in Figure A.3, using the linear quantizer (employed in the baselines Q-Diffusion and TFMQ-DM) causes slight changes in the image content, and at the 6-bit setting, the image becomes misaligned with the text prompt. However, with attention-aware quantization, the image is successfully preserved, matching the quality of the full-precision images even at the 6-bit setting.

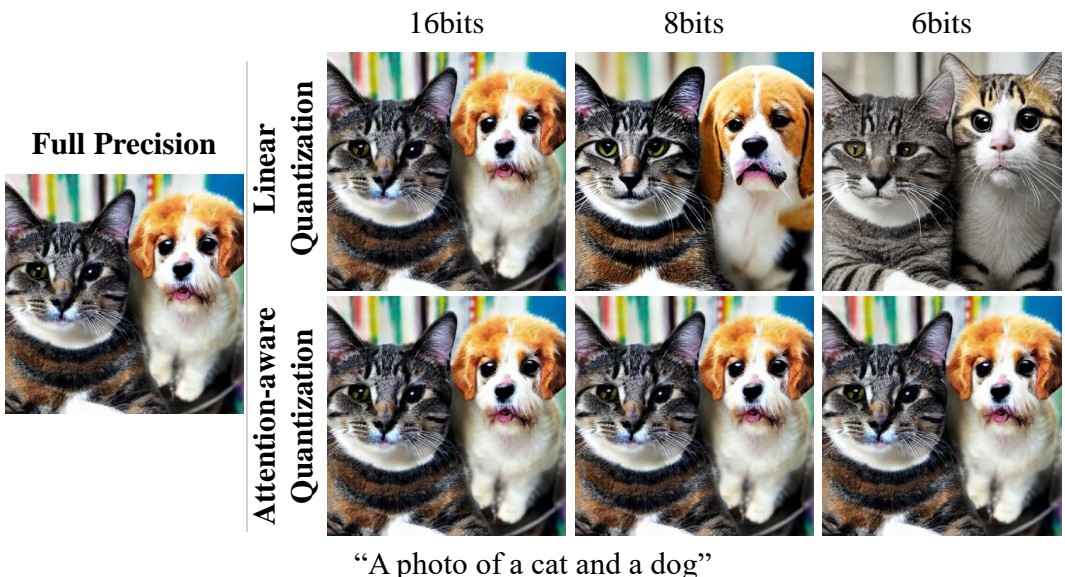

"A photo of a cat and a dog"

Figure A.3: **Qualitative comparison of attention score quantizer**

## G STATISTICS FOR ATTENTION SCORE DISTRIBUTION

To provide more detailed information about the attention score distribution, we calculated the statistics of attention scores and compared the distinct patterns between self-attention and cross-attention. We conducted experiments on the PartiPrompts dataset and collected the maximum value of each layer's attention scores. We transformed the maximum values using a base-2 logarithm ($\log_2$) and calculated the statistics.

As shown in Figure 5(b), the maximum values of cross-attention scores vary more dynamically than those of self-attention. According to the statistics of the maximum attention scores, the standard

| Statistic | Value |
|---|---|
| Std of cross-attention | 0.826 |
| Std of self-attention | 0.334 |
| Mean ratio of each layer's attention std | 3.210 |

Table A.3: **Statistics for attention score distribution.**

deviation of cross-attention is much larger than that of self-attention. The first two rows of Table A.3 present the standard deviation(std) of all maximum attention scores for each attention type. For a more accurate comparison, we calculated the standard deviation for each transformer layer and computed the mean of the ratios of these standard deviations. This confirms that the standard deviation of cross-attention is, on average, more than three times larger than that of self-attention.

## H    FULL VISUALIZATION OF ACTIVATION MATRIX

To better illustrate that outliers occur at a specific pixel (case 1) or channel (case 2), we visualized only the values around the indexes where outliers occur in Figure 4(b). Figure A.4 shows the visualization of full activation matrix.

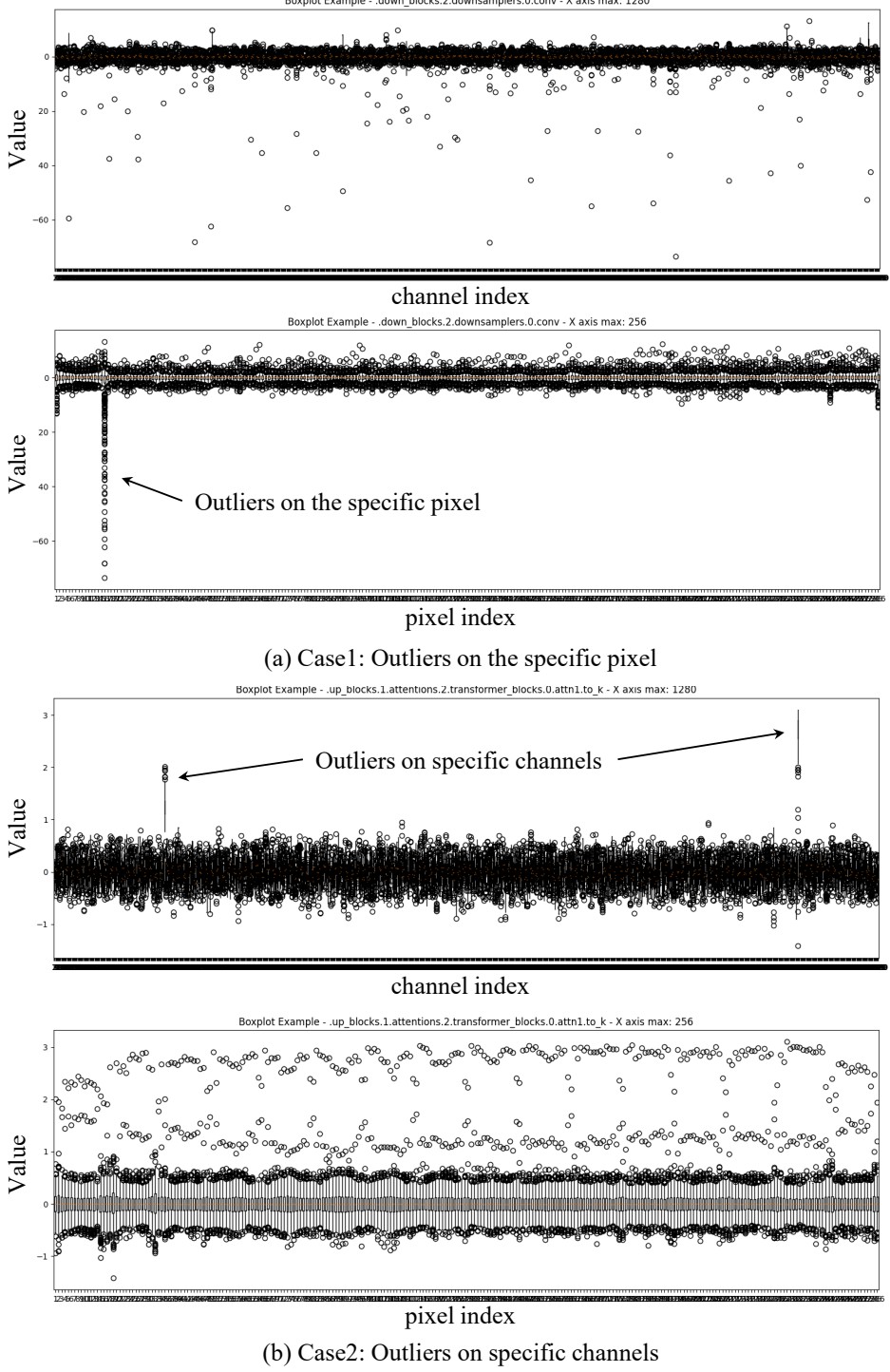

(a) Case1: Outliers on the specific pixel

(b) Case2: Outliers on specific channels

Figure A.4: **Visualization of full activation matrix in Figure 4(b)**

# I ADDITIONAL QUALITATIVE RESULTS

we provide more random samples from quantized models obtained using DGQ and TFMQ-DM. Results are shown in the figures below.

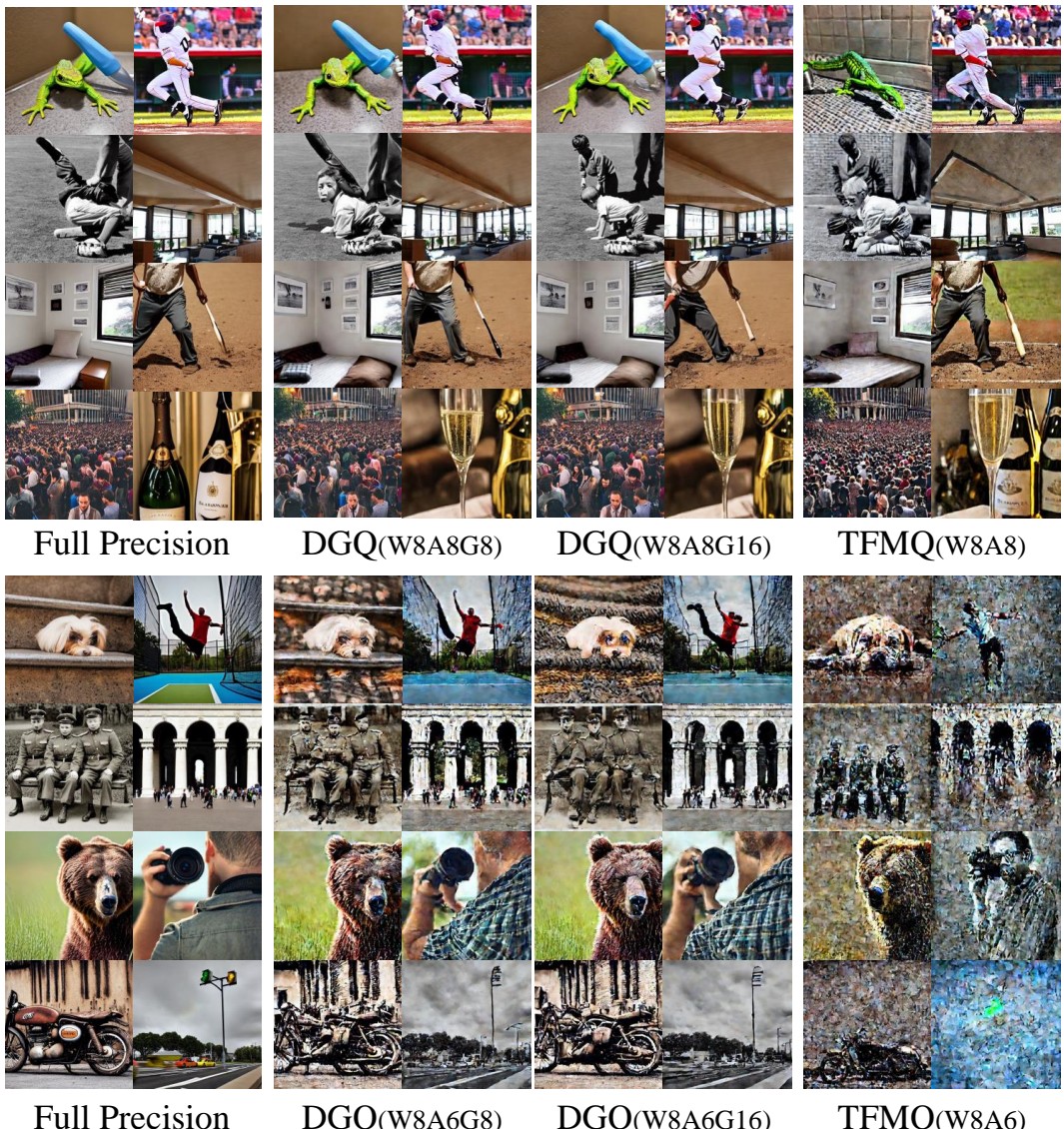

Figure A.5: **Additional qualitative results with the 8-bit weight, SDv1.4.** we randomly sampled the captions from the MS-COCO and generate images with them. WXAYGZ represents weights and activations with X and Y bits and a group size of Z.

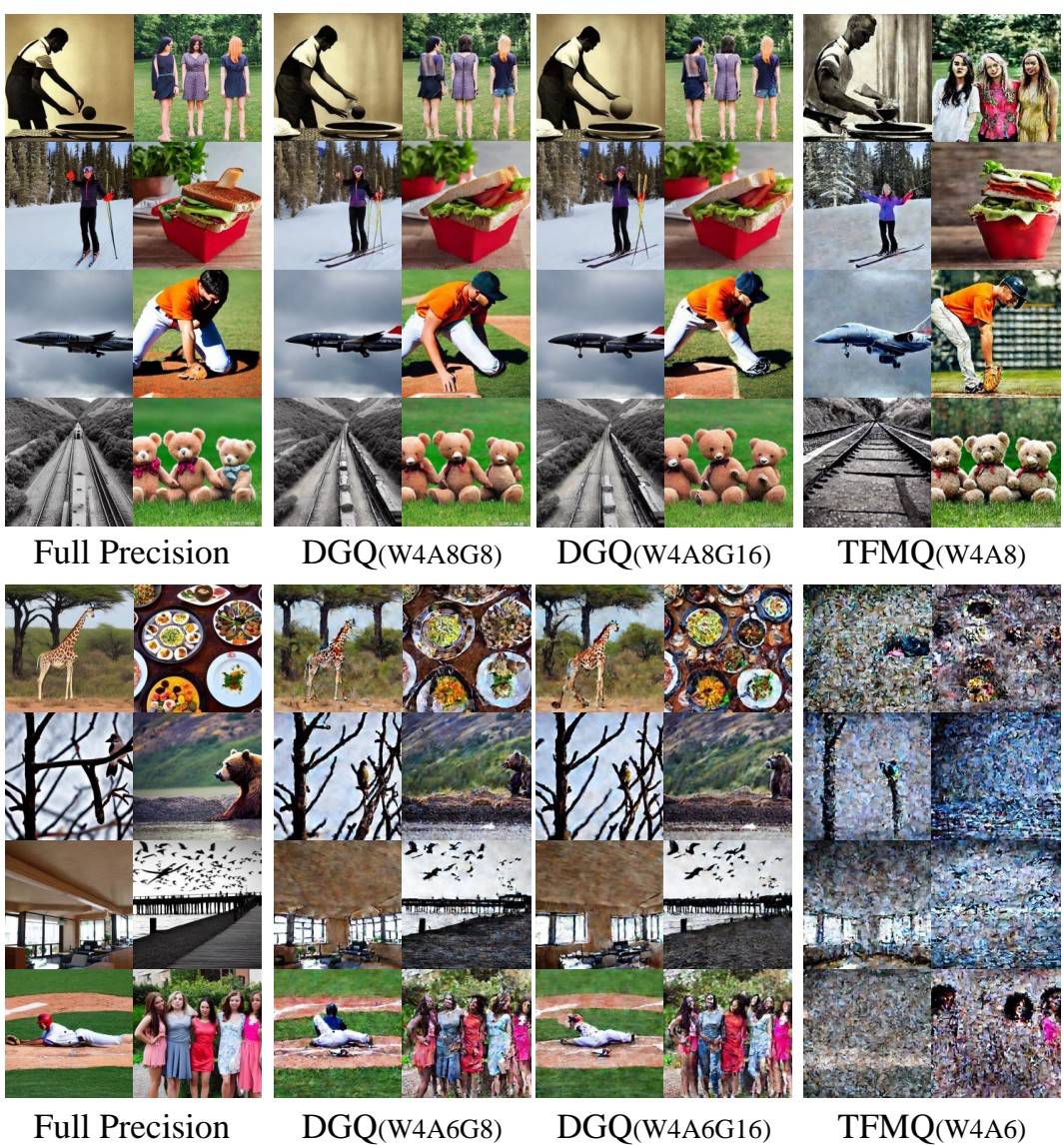

Figure A.6: **Additional qualitative results with the 4-bit weight, SDv1.4** we randomly sampled the captions from the MS-COCO and generate images with them. WXAYGZ represents weights and activations with X and Y bits and a group size of Z.

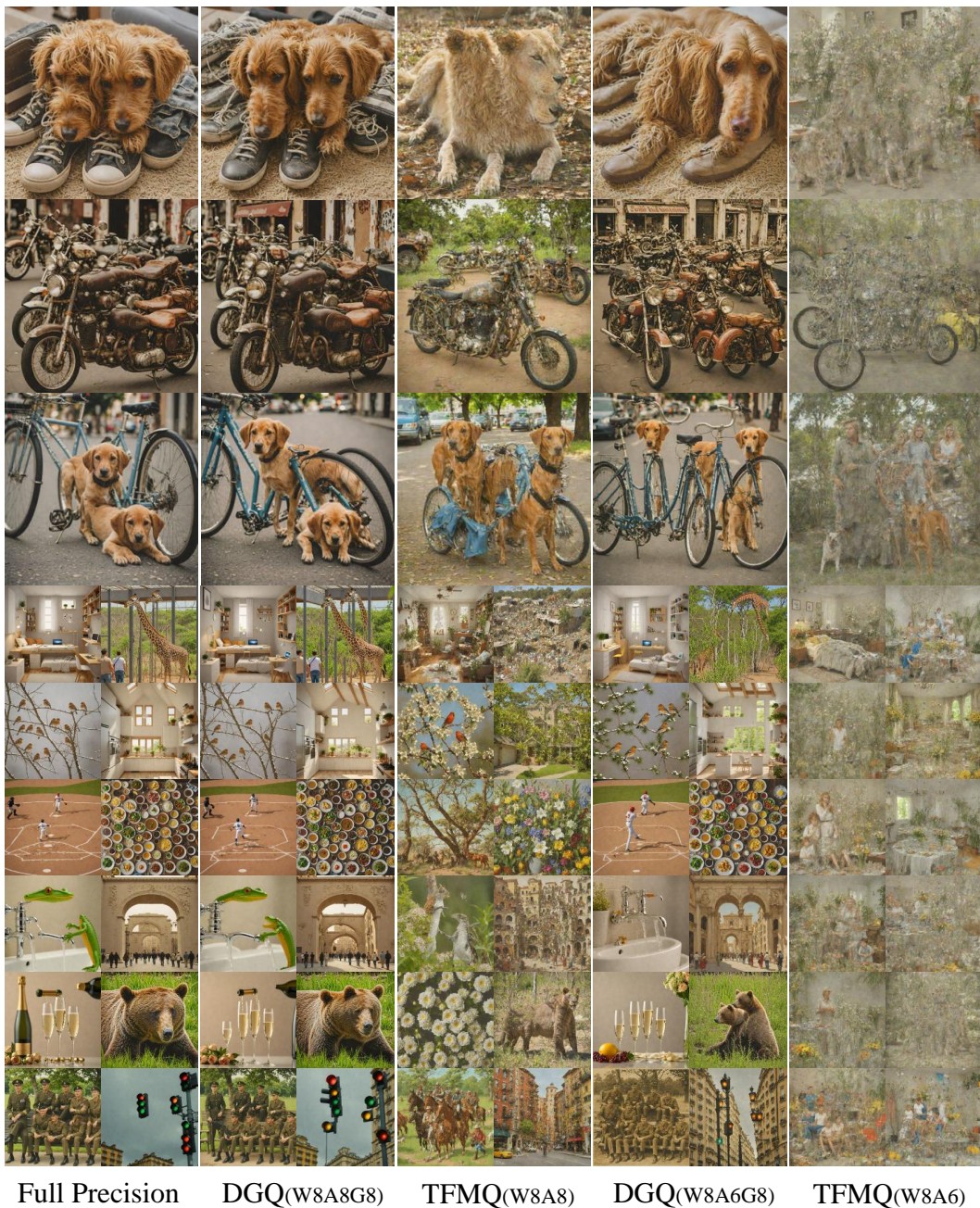

| Full Precision | DGQ(W8A8G8) | TFMQ(W8A8) | DGQ(W8A6G8) | TFMQ(W8A6) |

Figure A.7: **Additional qualitative results with the 8-bit weight, SDXL-Turbo.** we randomly sampled the captions from the MS-COCO and generate images with them. WXAYGZ represents weights and activations with X and Y bits and a group size of Z.

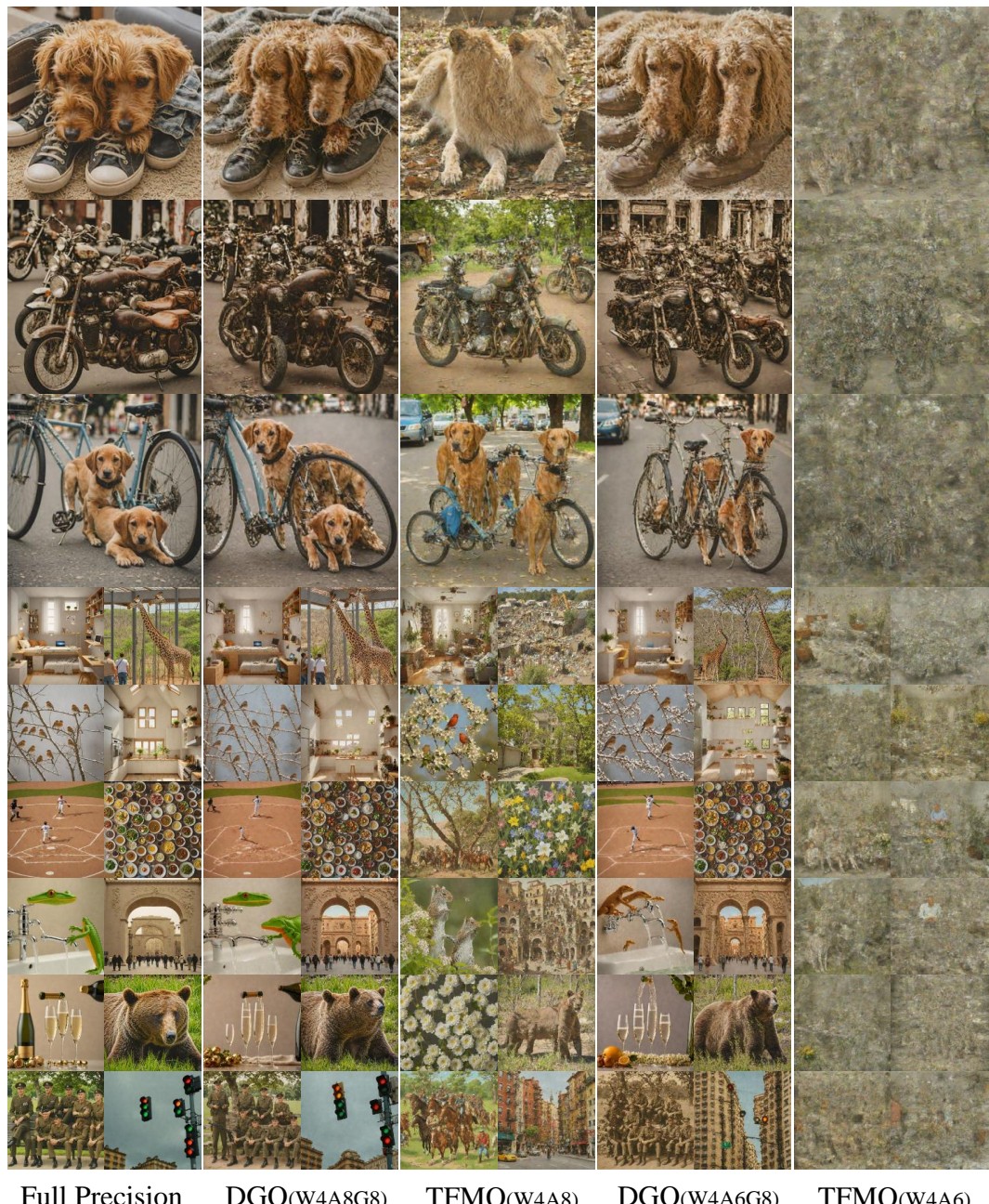

Full Precision     DGQ(W4A8G8)     TFMQ(W4A8)     DGQ(W4A6G8)     TFMQ(W4A6)

Figure A.8: **Additional qualitative results with the 4-bit weight, SDXL-Turbo.** we randomly sampled the captions from the MS-COCO and generate images with them. WXAYGZ represents weights and activations with X and Y bits and a group size of Z.

