# OpenReview forum: "DGQ: Distribution-Aware Group Quantization for Text-to-Image Diffusion Models"
_ICLR.cc/2025/Conference — ICLR 2025 Poster_

### Official Review · Reviewer_1KQn · 2024-10-28

**Soundness:** 3
**Presentation:** 4
**Contribution:** 2
**Rating:** 8
**Confidence:** 4

**Summary:**

This paper presents an analysis of feature distributions within layers and cross-attention scores in text-to-image (T2I) diffusion models. Building on these analysis, the authors introduce Distribution-aware Group Quantization (DGQ), a method specifically designed to address the distinct distribution patterns in T2I diffusion models. The proposed approach is validated across multiple datasets, demonstrating its effectiveness.

**Strengths:**

- The paper's motivation is clear and straightforward.

- The authors effectively identify and address outlier issues present in prior methods, providing a targeted method.

- The analysis of attention scores for <start> tokens is convincing, and the proposed method appears to address these challenges well.

**Weaknesses:**

- The authors validate the proposed method only on SDv1.4. These limited experiments restrict the ability to assess the generality of their approach. Additional analyses and experiments on other publicly available checkpoints (e.g., SD v2+, EDM, Imagen, etc) would strengthen the evaluation.

- Lines 127-129: "To the best of our knowledge, we are the first to achieve low-bit quantization (< 8-bit) on text-to-image diffusion models." However, [A] has previously proposed low-bit quantization for T2I diffusion models. While [A] and the submitted paper are concurrent works, a discussion of recent works would need for the future work.

- While the analysis and proposed method have novel elements, the approach relies heavily on existing methods (Group Quantization, logarithmic quantization). This reliance may worsen the novelty of the method in contrast to the insightful analysis.

[A] BitsFusion: 1.99 bits Weight Quantization of Diffusion Model, NeurIPS 2024

**Questions:**

The major concerns are in the weakness section, therefore I currently feel that this paper is borderline.
I have some other questions in below:

- The authors analyze the <start> token's attention score distribution; however, I am curious about the distributions for other special tokens such as <end> or <pad> tokens and a comparison discussion with <start> would find meaningful insight within the T2I diffusion model.

- As I understand it, the proposed method (Attention-aware quantization) applies full precision to the <start> token while quantizing other components to lower precision. Consequently, in Table 2, I would expect slight differences in memory req and BOPs compared to the baselines. However, the current paper shows identical values for these metrics. Could you clarify the difference?

- The evaluation includes FID and IS metrics for assessing image qualtiy. I think other image quality assesments (IQA, [A, B]) against full-precision and baseline models would provide practical insights and could enhance the evaluation for the both datasets.

[A] MANIQA: Multi-dimension Attention Network for No-Reference Image Quality Assessment, CVPRW 2022
[B] PromptIQA: Boosting the Performance and Generalization for No-Reference Image Quality Assessment via Prompts, ECCV 2024

---

> ### Author Response · Authors · 2024-11-19
> **Response for reviewer 1KQn (Part 1/2)**
>
> ### General Reply
>
> We sincerely appreciate your valuable and constructive feedback. Below, we provide our responses to your questions and comments. If there are any concerns that we have not adequately addressed, please let us know, and we will respond as soon as possible.
>
> ---
>
> ### **Weakness 1. Experiments on SDXL**
>
> > The authors validate the proposed method only on SDv1.4. These limited experiments restrict the ability to assess the generality of their approach. Additional analyses and experiments on other publicly available checkpoints (e.g., SD v2+, EDM, Imagen, etc) would strengthen the evaluation.
>
> Thank you for your valuable suggestion. Following your recommendation, we conducted experiments on the advanced model, SDXL-turbo. As in previous experiments, applying the proposed quantization method significantly improved performance compared to existing techniques. The quantitative results are presented in the table below.
>
> | Method         | Bits(W/A) | IS         | FID        | CLIP      |
> | -------------- | --------- | ---------- | ---------- | --------- |
> | Full Precision | 32/32     | 35.974     | 21.252     | 0.308     |
> | TFMQ           | 8/8       | 12.240     | 111.685    | 0.067     |
> | DGQ(#groups=8) | 8/8       | **34.789** | **22.455** | **0.299** |
> | TFMQ           | 8/6       | 4.271      | 163.020    | -0.002    |
> | DGQ(#groups=8) | 8/6       | **28.559** | **34.310** | **0.251** |
> | TFMQ           | 4/8       | 13.000     | 109.557    | 0.068     |
> | DGQ(#groups=8) | 4/8       | **28.332** | **29.218** | **0.289** |
> | TFMQ           | 4/6       | 1.985      | 270.446    | 0.022     |
> | DGQ(#groups=8) | 4/6       | **22.934** | **44.999** | **0.245** |
>
> Table. Quantitative Comparison of Quantization Methods on SDXL-Turbo(num_inference_step=4) with the MS-COCO 30k samples.
>
>
>
> While the **baseline fails to generate images even in the W8A8 setting (FID score exceeds 100), DGQ achieves outstanding performance in both FID and CLIP scores across all settings**: W8A8, W8A6, W4A8, and W4A6. Notably, in the W8A8 setting, DGQ performs on par with the full-precision model.
>
> We will include these experimental results in the final version. We sincerely thank you for providing such a valuable and impactful suggestion.
>
>
>
> ### **Weakness 2. Comparison with BitFusion**
>
> > Lines 127-129: "To the best of our knowledge, we are the first to achieve low-bit quantization (< 8-bit) on text-to-image diffusion models." However, [A] has previously proposed low-bit quantization for T2I diffusion models. While [A] and the submitted paper are concurrent works, a discussion of recent works would need for the future work.
>
> Thank you for pointing this out. BitFusion is conducted in a completely different setting(i.e., Quantization-Aware Training) from ours(i.e. Post-Training Quantization). BitFusion requires a huge dataset and significant computational cost to obtain a quantized model. In contrast, DGQ (Ours) is a model generated through PTQ(Post-Training Quantization) that does not require a dataset, requires only 64 prompts, and has a minimal computational cost. Specifically, according to the **BitFusion paper, their model was trained for 50K iterations with a batch size of 1024 using an internal dataset, utilizing 32 NVIDIA A100 GPUs.** On the other hand, **DGQ used only 64 sample prompts during the activation quantization process and was completed in about 20 minutes on just one RTX A6000** (based on Stable Diffusion v1.4 with 25 steps).
>
> Generally, models quantized through QAT have better performance compared to models quantized through PTQ. However, due to the need for a huge training dataset and high training costs, QAT is not practical. Therefore, recent quantization research for the large foundation models has been focused on PTQ(please refer to [1]). **DGQ is the first method to achieve low-bit quantization of text-to-image diffusion models without any additional fine-tuning (i.e., PTQ).**
>
> We will include this discussion and clarify the difference between the PTQ and QAT in the final version. Thank you for your valuable comments.
>
> [1] Zhu, Xunyu, et al. "A survey on model compression for large language models." *arXiv preprint arXiv:2308.07633* (2023).

---

> ### Author Response · Authors · 2024-11-19
> **Response for reviewer 1KQn (Part 2/2)**
>
> ### **Weakness 3. Reliance of methods.**
>
> > While the analysis and proposed method have novel elements, the approach relies heavily on existing methods (Group Quantization, logarithmic quantization). This reliance may worsen the novelty of the method in contrast to the insightful analysis.
>
> **Since quantization techniques need to be compatible with existing hardware, we prioritized developing methods that involve minimal modifications to existing methods.** Conversely, not relying on existing methods would require implementing new algorithms at the hardware level, significantly reducing practicality. For similar reasons, other research on quantization has also focused on enhancing performance while relying on existing methods.
>
> Additionally, as you mentioned, by examining the model from various perspectives, we were able to conduct insightful analyses. **Our methodology was tailored to align with this analysis. We believe that utilizing existing methods does not worsen the novelty.** Most importantly, our grouping strategy and the separation of <start> are new innovations that have never existed before, and without them, such outstanding performance would not have been possible.
>
> Thank you for your valuable comments.
>
>
>
> ---
>
> ### **Answers for Questions**
>
>
> **Question 1. The authors analyze the <start> token's attention score distribution; however, I am curious about the distributions for other special tokens such as <end> or <pad> tokens and a comparison discussion with <start> would find meaningful insight within the T2I diffusion model.**
>
> In the process of analyzing cross attention, we have already observed the attention scores of <pad> and <end> tokens. However, in the case of the two tokens, there were no special values in the attention scores. They usually had lower values than the attention scores assigned to text tokens or <start> tokens. Thank you for your question.
>
> **Question 2. As I understand it, the proposed method (Attention-aware quantization) applies full precision to the <start> token while quantizing other components to lower precision. Consequently, in Table 2, I would expect slight differences in memory req and BOPs compared to the baselines. However, the current paper shows identical values for these metrics. Could you clarify the difference?**
>
> Thanks for pointing it out. I will clarify the extra cost. As for memory requirements, Table 2 is true that it is a measurement of model size, and there is no change even if the <start> token is used with full precision. As for computation cost, it requires an additional 0.0237 TBOPs, compared to other baselines, which is a negligible number.
>
> **Question 3. The evaluation includes FID and IS metrics for assessing image quality. I think other image quality assessments (IQA, [A, B]) against full-precision and baseline models would provide practical insights and could enhance the evaluation for the both datasets.**
>
> Thank you for your valuable suggestion. I will evaluate the image quality using IQA methods. I will share the results once the evaluation is complete.

---

> > ### Author Response · Authors · 2024-11-27
> >
> > ### **Question 3. IQA evaluation results.**
> >
> > We have included the quantitative results of image quality evaluated using the IQA model (MANIQA) and a human preference reward model (Appendix C). The evaluation was conducted on 30,000 samples from the MS-COCO dataset. As shown in Table below(also in A.2 of revised PDF), DGQ significantly outperforms the baseline in almost all cases. Under 8-bit activation settings, while TFMQ achieves very slightly higher performance with MANIQA, DGQ achieves better results on the ImageReward model in all cases. Thank you for your valuable suggestion.
> >
> > | Method         | Bits(W/A) | MANIQA     | ImageReward |
> > | -------------- | --------- | ---------- | ----------- |
> > | Full Precision | 32/32     | 0.5525     | 0.1120      |
> > | TFMQ           | 8/8       | **0.5359** | -0.0375     |
> > | DGQ(#groups=8) | 8/8       | 0.5340     | **0.0668**  |
> > | TFMQ           | 8/6       | 0.3795     | -1.8643     |
> > | DGQ(#groups=8) | 8/6       | **0.4187** | **-0.2724** |
> > | TFMQ           | 4/8       | **0.5265** | -0.1222     |
> > | DGQ(#groups=8) | 4/8       | 0.5223     | **-0.0256** |
> > | TFMQ           | 4/6       | 0.3834     | -2.0587     |
> > | DGQ(#groups=8) | 4/6   | **0.4282** | **-0.4630** |

---

> ### Author Response · Authors · 2024-11-27
> **Gentle Reminder**
>
> Dear Reviewer 1KQn,
>
> Thank you for your valuable feedback and suggestions, which have greatly contributed to improving our paper. Based on your comments, we have conducted additional experiments on SDXL-turbo (Table 2), included a more detailed description of quantization-aware training and post-training quantization (Appendix D.2), and conducted evaluations using IQA and human preference reward models (Appendix C). All these updates have been included in the revised PDF.
>
> As the submission deadline for the revised paper is November 27, we kindly remind you to review our responses. We have made every effort to address your concerns thoroughly. We would greatly appreciate it if you could confirm whether our revisions have sufficiently resolved your questions and consider revisiting your score, or let us know if there are any further issues.
>
> Best regards,
> The Authors

---

> ### Comment · Reviewer_1KQn · 2024-11-29
>
> Most of my questions have been resolved. Thank you for the author's efforts in addressing the questions. After reading the other reviewers' comments and reviewing the responses, I raise the score to 8.

---

> > ### Author Response · Authors · 2024-12-02
> >
> > Thank you once again for reviewing our paper and providing valuable comments!

---

### Official Review · Reviewer_s9vr · 2024-10-30

**Soundness:** 3
**Presentation:** 3
**Contribution:** 3
**Rating:** 6
**Confidence:** 4

**Summary:**

The paper proposes a novel quantization method for diffusion models, named DGQ, which efficient reduce memory usage and computational costs compared to existing methods without additional finetuning. This approach aims to achieve high-quality generation while maintaining performance metrics. Experiments are conducted on MS-COCO and PartiPrompts using IS, FID and CLIP evaluation metrics.

**Strengths:**

1.It is interesting that the author finds activation outliers play a crucial role in determining image quality.
2.It is interesting that maintain text-image alignment in quantizing text-to-image diffusion models without requiring additional fine-tuning of weight quantization parameters
3.The logic of the work is clear, and the exploratory part of the experiments is rich in content

**Weaknesses:**

1.The conclusions in Figure 5(b) should provide statistical results.
2.This work presents several new matrix computations, including $D_d$. However, the authors fail to address the extra time overhead that these computations may incur.
3.In Table 2, the proposed method with groups set to 16 underperforms compared to groups set to 8 on some metrics (e.g., IS and FID). It would be beneficial to analyze the reasons behind this occurrence.

**Questions:**

Does excluding the <start> token mean dropping and zeroing it? Why does the author think that the peak near the value of 1 on the far right of Figure 5(a) corresponds to the <start> token, while other regions do not?

---

> ### Author Response · Authors · 2024-11-19
> **Response to reviewer s9vr**
>
> ### **General Reply**
>
> We sincerely appreciate your valuable and constructive feedback. Below, we provide our responses to your questions and comments. If there are any concerns that we have not adequately addressed, please let us know, and we will respond as soon as possible.
>
> ---
>
> ### **Weakness 2. Clarification of the extra overhead.**
>
> > This work presents several new matrix computations, including $D_d$. However, the authors fail to address the extra time overhead that these computations may incur.
>
> Thank you for pointing this out. The calculations of $D_d$, $S$, and $Z$ are performed during the activation quantizer calibration process and are not executed during the inference of the quantized model. Consequently, these calculations do not introduce any additional inference time overhead. Furthermore, the activation quantizer calibration was completed within 25 minutes on a single A6000 GPU (on Stable Diffusion v1.4 with 25 steps).
>
> In the case of separating the <start> token of attention-aware quantization, it does introduce additional overhead. However, the **corresponding BOPs are only 0.0237 TBOPs, which is significantly smaller than the quantized model’s BOPs (e.g., 51.45 TBOPs for the w8a8 setting) and can therefore be considered negligible.**
>
>
>
> ### **Weakness 3. Analysis on group size.**
>
> > In Table 2, the proposed method with groups set to 16 underperforms compared to groups set to 8 on some metrics (e.g., IS and FID). It would be beneficial to analyze the reasons behind this occurrence.
>
> Thank you for your suggestion. We address this topic in the **"Effects of Grouping Strategy." of the Ablation study** section.
>
> Increasing the group size is not always beneficial. As shown in Table 3(b), for the 8-bit activation setting, group sizes of 2 and 4 achieved the best performance in terms of FID and CLIP scores, respectively. However, as shown in Table A.1 in the appendix, for the 6-bit activation setting, performance improved as the group size increased.
>
> We interpret this phenomenon as follows. **For 8 bits, even with group sizes of 2 or 4, the quantization scale is already sufficiently small, so adding more groups provides minimal performance gain and can lead to overfitting.** In contrast, for 6 bits, the quantization scale is not small enough, allowing for additional performance gains even when the group size is set to 16.
>
>
>
> ---
>
> ### **Weakness 1. statistical results.**
>
> > The conclusions in Figure 5(b) should provide statistical results.
>
> Thank you for pointing this out. We will prepare the statistical results and once completed, we will write the comments along with the results.
>
>
>
> ---
> ### **Answers for questions.**
>
>
> **Question 1. Does excluding the <start> token mean dropping and zeroing it?**
>
> No, ‘excluding <start> token’ in figure 5(a) means excluding the attention scores corresponding to <start> token.
>
>
>
> **Question 2. Why does the author think that the peak near the value of 1 on the far right of Figure 5(a) corresponds to the <start> token, while other regions do not?**
>
> We noticed that the many attention scores near 1 correspond to the <start> token by observing the actual value of the activation matrix. However, it does not mean that the scores corresponding to the <start> token are always near 1. As mentioned on lines 292-294, attention scores of background pixels are often concentrated on <start> token($\simeq 0.99$), and it leads to the presence of peak near 1, but those of foreground pixels are not concentrated on <start> token.
>
> Thank you for your valuable questions.

---

> > ### Author Response · Authors · 2024-11-27
> >
> > ### **Weakness 1: Statistical Results**
> >
> > We have provided statistics on the distribution of attention scores. The experiment was conducted on the PartiPrompts dataset, where we collected the maximum value of each layer's attention scores. Since our method uses a logarithmic quantizer, we transformed the maximum values to base-2 logarithms and calculated the statistics.
> >
> > As shown in the table below, the standard deviation of cross-attention (0.826) is much larger than that of self-attention (0.334), which means the maximum values of the cross-attention score distribution vary more dynamically than those of self-attention.
> >
> > For a more accurate comparison, we calculated the standard deviation for each transformer layer and computed the ratio between those of cross-attention and self-attention. On average, the standard deviation of the cross-attention scores' maximum values is more than three times larger than that of self-attention (3.210).
> >
> > This result is included in the revised PDF (Appendix G). Thank you for your valuable suggestion.
> >
> >
> >
> > | **Statistics**                               | **Values** |
> > | ---------------------------------------- | ------ |
> > | Std of cross-attention                   | 0.826  |
> > | Std of self-attention                    | 0.334  |
> > | Mean ratio of each layer's attention std | 3.210  |

---

> ### Author Response · Authors · 2024-11-27
> **Gentle Reminder**
>
> Dear Reviewer s9vr,
>
> Thank you for your valuable feedback and suggestions, which have greatly contributed to improving our paper. Based on your comments, we have provided the statistical results supporting the conclusions in Figure 5(b) (Appendix G) and have clarified the extra overhead and the effect of the group size. Thank you for pointing out this important aspect.
>
> Furthermore, we have made several additional updates in the revised PDF, including experiments on SDXL-turbo.
>
> As the submission deadline for the revised paper is November 27, we kindly remind you to review our responses. We have made every effort to address your concerns thoroughly. We would greatly appreciate it if you could confirm whether our revisions have sufficiently resolved your questions, or let us know if there are any further issues.
>
> Best regards,
>
> The Authors

---

> ### Author Response · Authors · 2024-12-02
>
> Dear Reviewer s9vr,
>
> As the discussion phase is nearing its end, we kindly ask you to let us know if our response has addressed your concerns or if you require any additional information.
>
> We once again thank you for the time you've dedicated to reviewing our paper and for your valuable feedback.
>
> Best regards,
>
> The Authors

---

### Official Review · Reviewer_Hfso · 2024-10-31

**Soundness:** 3
**Presentation:** 3
**Contribution:** 3
**Rating:** 6
**Confidence:** 4

**Summary:**

This paper proposes Distribution-aware Group Quantization (DGQ), which identifies and adaptively manages pixel-wise and channel-wise outliers to preserve image quality. Additionally, a prompt-specific logarithmic quantization strategy is introduced to enhance text-image alignment. Experiments conducted on the MS-COCO dataset demonstrate the effectiveness of DGQ.

**Strengths:**

This paper exhibits several strengths:

1. The analysis of key characteristics necessary for maintaining model performance during the quantization process is scientific and persuasive.

2. The proposed group-wise quantization strategy, based on the distribution of activations, is both promising and innovative.

**Weaknesses:**

This paper exhibits several Weaknesses:

1.	The figures are confusing and difficult to understand.

     a)   How are issues like "drops outlier," "high quantization error," and "high overhead" reflected in Fig. 3? What specific phenomenon indicates "drops outlier"?

    b)    What does the horizontal axis in Fig. 4(b) represent? In channel-wise boxplot, why is the horizontal axis range of case1 from 0 to 900, while for case 2 it ranges from 1100 to 1190? What do "specific pixel" and "specific channel" refer to?

2.	The distinctions between group-wise quantization, layer-wise quantization, and channel-wise quantization are somewhat abstract and challenging to grasp. It would be beneficial for the authors to include a figure that illustrates the differences in their principles.

3.	There is a lack of qualitative and quantitative comparisons with the latest research, such as MixDQ[1].

4.	The base model used is outdated and singular. It is recommended to include experiments with more recent Text-to-Image models (e.g., SDXL, PixArt, SD3, or Flux) to validate the generalization of DGQ.


[1] MixDQ: Memory-Efficient Few-Step Text-to-Image Diffusion Models with Metric-Decoupled Mixed Precision Quantization

**Questions:**

1.	How can the quantized model in Table 2 outperform the original model? What mechanisms allow the quantization process to enhance performance?

2.	The meaning of W8A8 should be explained upon its first mention. The article lacks clarity on the notation WXAX, and similarly, the meaning of WXAXGX in the supplementary materials is not explained.

3.	It is recommended to add brackets around the feature indices in Formula 9 for improved clarity, such as A_{[:,0]} V_{[0,:]}.

4.	Why does reducing the bits of activation result in significant performance loss?

**Details Of Ethics Concerns:**

Nan

---

> ### Author Response · Authors · 2024-11-19
> **Response to reviewer Hfso (Part 1/2)**
>
> ### **General Reply**
>
> We sincerely appreciate your valuable and constructive feedback. Below, we provide our responses to your questions and comments. If there are any concerns that we have not adequately addressed, please let us know, and we will respond as soon as possible.
>
> ---
>
> ### **Weakness 1. Clarification of figures.**
>
> **a) How are issues like "drops outlier," "high quantization error," and "high overhead" reflected in Fig. 3? What specific phenomenon indicates "drops outlier"?**
>
> Thank you for your question. In figure 3, the gray lines represent the quantized values. When the full precision activation values (boxplot) are quantized, they are converted to the nearest quantized values (gray dotted lines).
>
> In this figure, **“drops outlier”** is described by the absence of a gray dotted line near the outlier value(red boxplot), indicating that the outlier is not preserved. **“High quantization error”** is represented by the wide spacing between the gray dotted lines, illustrating that the quantization levels are high(low bit-width). **“High overhead”** refers to the issue that arises when using channel-wise quantization.
>
> Therefore, in the case of group-wise quantization shown in the Figure, there is a gray dotted line even near the outlier (preserving the outlier), and the spacing between the gray dotted lines is relatively narrow (indicating low quantization error). Moreover, since group-wise quantization has much less overhead compared to channel-wise quantization, it is marked as having low overhead.
>
>
>
> **b.1) What does the horizontal axis in Fig. 4(b) represent?**
>
> The horizontal axis represents the pixel index (left) and the channel index (right). Each case visualizes the values of the same activation matrix (channel × pixel) along the pixel axis and the channel axis, respectively. In the final version, we will label the axes of each graph with 'pixel index' and 'channel index' below them.
>
> **b.2) What do "specific pixel" and "specific channel" refer to?**
>
> For the “specific pixel” case, it refers to the pixel where an outlier occurs between indices 10 and 20 in the pixel-wise boxplot of Case 1. The “specific channel” refers to the channel where an outlier occurs between indices 1150 and 1560 in the channel-wise boxplot of Case 2.
>
> **b.3) In channel-wise boxplot, why is the horizontal axis range of case1 from 0 to 900, while for case 2 it ranges from 1100 to 1190?**
>
> To better illustrate that outliers occur at a specific pixel (case 1) or channel (case 2), we visualized only the values around the indexes where outliers occur. In the case of the matrix in case 2, the original dimension is 1280, but visualizing all indices from 0 to 1280 would make each boxplot too small, making it difficult to observe the occurrence of outliers. Therefore, we only visualized the range of indexes around where outliers occur. We will include images showing the full channels and pixels in the Appendix.
>
>
>
> ### **Weakness 2. Figure for quantization granularity.**
>
> >  The distinctions between group-wise quantization, layer-wise quantization, and channel-wise quantization are somewhat abstract and challenging to grasp. It would be beneficial for the authors to include a figure that illustrates the differences in their principles.
>
> Thank you for pointing out that the differences between group-wise quantization, layer-wise quantization, and channel-wise quantization can be hard to understand. We will add a figure to explain these concepts more clearly.
>
> For an activation matrix $m \times n$, layer-wise quantization uses a single quantizer for the entire matrix, group-wise quantization uses a different quantizer for each channel group, and channel-wise quantization uses a different quantizer for each activation channel.
>
>
>
> ### **Weakness 3. Comparison with MixDQ.**
>
> > There is a lack of qualitative and quantitative comparisons with the latest research, such as MixDQ [1].
>
> **MixDQ[1] focuses on mixed-precision quantization.** It proposes a strategy to determine how many bits to allocate to the quantizer of each layer, but it does not propose how to perform quantization for the layer itself, meaning it is not about the quantizer's algorithm.
>
> **Since mixed-precision quantization utilizes additional hardware resources for supporting multiple precisions compared to fixed-precision, we believe it is not an apples-to-apples comparison with our proposed method**. However, if MixDQ were integrated into our approach, further performance improvements could be expected. Thank you for pointing this out.

---

> ### Author Response · Authors · 2024-11-19
> **Response to reviewer Hfso (Part 2/2)**
>
> ### **Weakness 4. Experiments on SDXL**
>
> > The base model used is outdated and singular. It is recommended to include experiments with more recent Text-to-Image models (e.g., SDXL, PixArt, SD3, or Flux) to validate the generalization of DGQ.
>
> Thank you for your valuable suggestion. Following your recommendation, we conducted experiments on the advanced model, SDXL-turbo. As in previous experiments, applying the proposed quantization method significantly improved performance compared to existing techniques. The quantitative results are presented in the table below.
>
> | Method         | Bits(W/A) | IS         | FID        | CLIP      |
> | -------------- | --------- | ---------- | ---------- | --------- |
> | Full Precision | 32/32     | 35.974     | 21.252     | 0.308     |
> | TFMQ           | 8/8       | 12.240     | 111.685    | 0.067     |
> | DGQ(#groups=8) | 8/8       | **34.789** | **22.455** | **0.299** |
> | TFMQ           | 8/6       | 4.271      | 163.020    | -0.002    |
> | DGQ(#groups=8) | 8/6       | **28.559** | **34.310** | **0.251** |
> | TFMQ           | 4/8       | 13.000     | 109.557    | 0.068     |
> | DGQ(#groups=8) | 4/8       | **28.332** | **29.218** | **0.289** |
> | TFMQ           | 4/6       | 1.985      | 270.446    | 0.022     |
> | DGQ(#groups=8) | 4/6       | **22.934** | **44.999** | **0.245** |
>
> Table. Quantitative Comparison of Quantization Methods on SDXL-Turbo(num_inference_step=4) with the MS-COCO 30k samples.
>
>
>
> While the **baseline fails to generate images even in the W8A8 setting (FID score exceeds 100), DGQ achieves outstanding performance in both FID and CLIP scores across all settings**: W8A8, W8A6, W4A8, and W4A6. Notably, in the W8A8 setting, DGQ performs on par with the full-precision model.
>
> We will include these experimental results in the final version. We sincerely thank you for providing such a valuable and impactful suggestion.
>
> ---
>
> ### **Answers for Questions**
>
> **Question 1. How can the quantized model in Table 2 outperform the original model? What mechanisms allow the quantization process to enhance performance?**
>
> Thank you for your question. Our method utilizes BRECQ [2], consistent with the baseline, during the weight quantization process. BRECQ approximates the loss degradation that occurs during the quantization process and quantizes the model to minimize it. We speculate that this process can achieve better performance than the original full-precision model.
>
> In various model quantization methods that leverage BRECQ (e.g., TFMQ-DM, PTQ4SAM [3]; please refer to the experiment table in the original paper), it is not uncommon to achieve higher performance than the full-precision model. For further details, please refer to the BRECQ paper.
>
>
>
> **Question 2. The meaning of W8A8 should be explained upon its first mention. The article lacks clarity on the notation WXAX, and similarly, the meaning of WXAXGX in the supplementary materials is not explained.**
>
> Thank you for pointing this out. WXAY represents weights and activations with X and Y bits, respectively, while WXAYGZ represents weights and activations with X and Y bits and a group size of Z. I will include this explanation in the final version.
>
>
>
> **Question 3. It is recommended to add brackets around the feature indices in Formula 9 for improved clarity, such as A_{[:,0]} V_{[0,:]}.**
>
> Thank you for your valuable suggestion. We will update the equations in the final version.
>
>
>
> **Question 4. Why does reducing the bits of activation result in significant performance loss?**
>
> Thanks for your question. As shown in lines 94-102 of our paper, quantizing activations with small bit widths leads to performance degradation as it is impossible to simultaneously preserve outliers and reduce quantization errors.
>
> Additionally, weights commonly include a regularization loss in the objective function during training, which reduces the likelihood of outliers and makes them easier to quantize. In contrast, activations typically lack regularization, leading to irregular patterns (e.g., outliers) and making them harder to quantize.
>
>
> ---
>
> [1]Zhao, Tianchen, et al. "MixDQ: Memory-Efficient Few-Step Text-to-Image Diffusion Models with Metric-Decoupled Mixed Precision Quantization." *arXiv preprint arXiv:2405.17873* (2024).
>
> [2] Li, Yuhang, et al. "Brecq: Pushing the limit of post-training quantization by block reconstruction." *arXiv preprint arXiv:2102.05426* (2021).
>
> [3] Lv, Chengtao, et al. "PTQ4SAM: Post-Training Quantization for Segment Anything." Proceedings of the IEEE/CVF Conference on Computer Vision and Pattern Recognition. 2024.

---

> > ### Comment · Reviewer_Hfso · 2024-11-19
> > **Response to authers**
> >
> > Thank you for your reply. ICLR should allow resubmission of the revised PDF. I recommend that the authors submit the revised PDF so that the reviewers can easily assess the final presentation of the paper

---

> > > ### Author Response · Authors · 2024-11-19
> > >
> > > We are currently preparing a revised PDF. We will notify you by leaving a comment once the revision is complete. Thank you.

---

> > > ### Author Response · Authors · 2024-11-24
> > >
> > > We have uploaded the revised PDF. You can check it now!

---

> ### Author Response · Authors · 2024-11-27
> **Gentle Reminder**
>
> Dear Reviewer Hfso,
>
> Thank you for your valuable feedback and suggestions, which have greatly contributed to improving our paper. Based on your comments, we have updated Figure 4(b) and added the full visualization in Appendix H. We have also included a figure and description of quantization granularity in Appendix E, and added experimental results on SDXL-turbo. We are very grateful for your thorough review and for pointing out even the smallest details. All these updates are included in the revised PDF.
>
> As the submission deadline for the revised paper is November 27, we kindly remind you to review our responses. We have made every effort to address your concerns thoroughly. We would greatly appreciate it if you could confirm whether our revisions have sufficiently resolved your questions, or let us know if there are any further issues.
>
> Best regards,
>
> The Authors

---

> ### Author Response · Authors · 2024-12-02
>
> Dear Reviewer Hfso,
>
> As the discussion phase is nearing its end, we kindly ask you to let us know if our response has addressed your concerns or if you require any additional information.
>
> We once again thank you for the time you've dedicated to reviewing our paper and for your valuable feedback.
>
> Best regards,
>
> The Authors

---

### Official Review · Reviewer_83s4 · 2024-11-01

**Soundness:** 4
**Presentation:** 3
**Contribution:** 4
**Rating:** 8
**Confidence:** 3

**Summary:**

This paper deeply explores the unique distribution patterns of activations and cross-attention scores in text-to-image diffusion models. Based on the analysis of patterns, the authors propose a method called Distribution-aware Group Quantization, which is composed of outlier-preserving group quantization for activations and a new quantizer for cross-attention scores. Experimental results show that the proposed method outperforms the state-of-the-art methods and achieves under 8-bit quantization for the first time.

**Strengths:**

1. This paper is well organized and the writing is good.
2. The in-depth analysis of the activation and cross-attention in the text-to-image diffusion models is interesting and may benefit the community.
3. The experiments are extensive and solid, showing the effectiveness of the proposed approach. The improvement is significant.
4. The idea is reasonable and easy to follow.

**Weaknesses:**

1. Typos from line.326 to line.327: these lines contain repeated sentences.
2. The number of generated samples to compute FID and IS is not provided from line.403 to line.410.
3. The baseline model SD 1.4 may be out-of-date now. It would be better if results on more advanced models like SDXL were provided.

**Questions:**

Please refer to the weakness part.

---

> ### Author Response · Authors · 2024-11-19
> **Response to reviewer 83s4**
>
> ### **General Reply**
>
> We sincerely appreciate your valuable and constructive feedback. Below, we provide our responses to your questions and comments. If there are any concerns that we have not adequately addressed, please let us know, and we will respond as soon as possible.
>
> ---
>
> ### **Weakness 3. Experiments on SDXL**
>
> > The baseline model SD 1.4 may be out-of-date now. It would be better if results on more advanced models like SDXL were provided.
>
> Thank you for your valuable suggestion. Following your recommendation, we conducted experiments on the advanced model, SDXL-turbo. As in previous experiments, applying the proposed quantization method significantly improved performance compared to existing techniques. The quantitative results are presented in the table below.
>
> | Method         | Bits(W/A) | IS         | FID        | CLIP      |
> | -------------- | --------- | ---------- | ---------- | --------- |
> | Full Precision | 32/32     | 35.974     | 21.252     | 0.308     |
> | TFMQ           | 8/8       | 12.240     | 111.685    | 0.067     |
> | DGQ(#groups=8) | 8/8       | **34.789** | **22.455** | **0.299** |
> | TFMQ           | 8/6       | 4.271      | 163.020    | -0.002    |
> | DGQ(#groups=8) | 8/6       | **28.559** | **34.310** | **0.251** |
> | TFMQ           | 4/8       | 13.000     | 109.557    | 0.068     |
> | DGQ(#groups=8) | 4/8       | **28.332** | **29.218** | **0.289** |
> | TFMQ           | 4/6       | 1.985      | 270.446    | 0.022     |
> | DGQ(#groups=8) | 4/6       | **22.934** | **44.999** | **0.245** |
>
> Table. Quantitative Comparison of Quantization Methods on SDXL-Turbo(num_inference_step=4) with the MS-COCO 30k samples.
>
>
>
> While the **baseline fails to generate images even in the W8A8 setting (FID score exceeds 100), DGQ achieves outstanding performance in both FID and CLIP scores across all settings: W8A8, W8A6, W4A8, and W4A6.** Notably, in the W8A8 setting, DGQ performs on par with the full-precision model.
>
> We will include these experimental results in the final version. We sincerely thank you for providing such a valuable and impactful suggestion.
>
>
>
> ### **Weakness 1 & 2. Typos and Implementation Details**
>
> >  Typos from line.326 to line.327: these lines contain repeated sentences.
>
> >  The number of generated samples to compute FID and IS is not provided from line.403 to line.410.
>
> Thank you for pointing this out. The number of samples used to measure FID and IS is 30K. We will remove the duplicate sentence and add the information about the number of samples in the final version.

---

> > ### Comment · Reviewer_83s4 · 2024-11-27
> >
> > Thanks for your responses. My concerns have been addressed.

---

> ### Author Response · Authors · 2024-11-27
> **Gentle Reminder**
>
> Dear Reviewer 83s4,
>
> Thank you for your valuable feedback and suggestions, which have greatly contributed to improving our paper. Based on your comments, we have conducted the experiments and provided the experimental results on SDXL-turbo (Table 2). We are very grateful for your thorough review and for pointing out even the smallest details(typos). All these updates are included in the revised PDF.
>
> As the submission deadline for the revised paper is November 27, we kindly remind you to review our responses. We have made every effort to address your concerns thoroughly. We would greatly appreciate it if you could confirm whether our revisions have sufficiently resolved your questions, or let us know if there are any further issues.
>
> Best regards,
>
> The Authors

---

### Official Review · Reviewer_qNA8 · 2024-11-05

**Soundness:** 3
**Presentation:** 2
**Contribution:** 2
**Rating:** 6
**Confidence:** 3

**Summary:**

This paper addresses the challenges of quantizing text-to-image diffusion models, which suffer from high memory and computational costs. The authors propose a method called Distribution-aware Group Quantization (DGQ), which handles activation outliers and applies prompt-specific quantization scales to preserve image quality and text-image alignment.

**Strengths:**

1. The writing is clear and easy to understand.

2. The experiments are thorough, and the analysis of outliers value offers valuable insights.

**Weaknesses:**

1. The evaluation of inference performance is lacking.

2. In line 417, it states, “we set all attention score quantizer bits to match the activation bits.” Could this direct approach negatively impact performance? What is the FLOP cost of setting the attention score quantizer bits to 16?

3. As shown in Figure A.3, there is still a significant performance drop when activation quantization bits are below 8.

**Questions:**

Refering to Weakness.

---

> ### Author Response · Authors · 2024-11-19
> **Response to reviewer qNA8**
>
> ### **General Reply**
>
> We sincerely appreciate your valuable and constructive feedback. Below, we provide our responses to your questions and comments. If there are any concerns that we have not adequately addressed, please let us know, and we will respond as soon as possible.
>
> ---
>
> ### **Weakness 2. Impact of attention score quantizer & Computational cost**
>
> > In line 417, it states, “we set all attention score quantizer bits to match the activation bits.” Could this direct approach negatively impact performance? What is the FLOP cost of setting the attention score quantizer bits to 16?
>
> **Q. Does setting attention score quantizer bit-width to lower value(activation’s bit-width) negatively impact performance?**
>
> **A. With linear quantizer, Yes. But, with DGQ, No.**
>
> Thank you for your question. When using a linear quantizer as the attention score quantizer, reducing the bit-width directly causes performance degradation. We have conducted an experiment by adjusting only the bit-width of the attention score quantizer in cross-attention, while keeping all other parts of the model in full precision. The result shows that as the bit-width decreases, image content changes significantly, and text-image alignment deteriorates noticeably. However, with our proposed method (DGQ), performance degradation is minimal. We will include qualitative results from these experiments in the final version.
>
>
>
> **Q. What is the FLOP cost of setting the attention score quantizer bits to 16?**
>
> **A. The FLOPs of the multiplication between the attention score and value is 63.03G, which is 7.84% of the full model(SDv1.4)'s FLOPs** (the full model's FLOPs is 803.96G = 823.26TBOPs, note that in the submitted paper, the BOPs calculation did not include the attention mechanism's BOPs, so the value of BOPs differ slightly from main results(Table 2); we will update this in the final version.).
>
> While the computation cost of the attention score multiplication in SDv1.4 is not a large proportion compared to the entire model, setting bit-width to 16 bits is not a scalable solution when considering the future direction of models. Considering the increasing use of Diffusion Transformers and the fact that the computational cost of the attention mechanism increases exponentially with higher image resolutions, it is necessary to set the attention scores to lower bit precision as well.
>
>
>
>
> ### **Weakness 3. Qualitative results**
>
> > As shown in Figure A.3, there is still a significant performance drop when activation quantization bits are below 8.
>
> Thank you for pointing this out. Although our method could not achieve the same performance as the full-precision model in the w4a6 setting, **it shows a significant performance improvement compared to the other methods.** Notably, while images generated by the other methods are unrecognizable in terms of their original content, our method preserves sufficient semantic information to make the images clearly interpretable.
>
> **The significance of our work lies in pushing the boundary of activation quantization with minimal changes to the quantization method without utilizing additional resources.** If combined with techniques such as fine-tuning or mixed precision, our method could achieve even greater performance improvements.
>
> Furthermore, as shown in the experimental results on SDXL-turbo (Table below), **the baseline completely fails to generate images even under a 8-bit activation setting, while our DGQ produces high-quality images on 6-bits.** We will include qualitative results in the final version.
>
> | Method         | Bits(W/A) | IS         | FID        | CLIP      |
> | -------------- | --------- | ---------- | ---------- | --------- |
> | Full Precision | 32/32     | 35.974     | 21.252     | 0.308     |
> | TFMQ           | 8/8       | 12.240     | 111.685    | 0.067     |
> | DGQ(#groups=8) | 8/8       | **34.789** | **22.455** | **0.299** |
> | TFMQ           | 8/6       | 4.271      | 163.020    | -0.002    |
> | DGQ(#groups=8) | 8/6       | **28.559** | **34.310** | **0.251** |
> | TFMQ           | 4/8       | 13.000     | 109.557    | 0.068     |
> | DGQ(#groups=8) | 4/8       | **28.332** | **29.218** | **0.289** |
> | TFMQ           | 4/6       | 1.985      | 270.446    | 0.022     |
> | DGQ(#groups=8) | 4/6       | **22.934** | **44.999** | **0.245** |
>
> Table. Quantitative Comparison of Quantization Methods on SDXL-Turbo(num_inference_step=4) with the MS-COCO 30k samples.
>
>
>
> ---
>
> ### **Weakness 1. Inference performance**
>
> > The evaluation of inference performance is lacking.
>
> Thank you for bringing this to our attention. We are currently working on the code to evaluate inference performance. Please note that implementing operations using CUTLASS is a time-intensive process, and we are trying hard to complete it. If we obtain the results, we will share them in this comment.

---

> > ### Comment · Reviewer_qNA8 · 2024-11-26
> >
> > Thank you for your reply. After thorough consideration, I believe the previous score sufficiently represents the quality of the article, so I have decided to maintain the current score.

---

> > > ### Author Response · Authors · 2024-11-27
> > >
> > > Thank you for taking the time to review our paper and for providing valuable comments!

---

### Author Response · Authors · 2024-11-24

Dear reviewers,

We sincerely thank the reviewers for their praise regarding the writing of our paper(**qNA8, 83s4, 1KQn**). We also appreciate the recognition of our analysis and experiments by all reviewers (**qNA8, 83s4, Hfso, s9vr, 1KQn**), and their evaluation that our proposed method and ideas are reasonable, clear, promising, innovative, or effective(**83s4, HFso, s9vr**).

---
Furthermore, we would like to thank you once again for taking the time to review our paper and for providing constructive and valuable comments to improve it. We have updated the manuscript by addressing the points you raised. The revised parts are highlighted in blue. The main changes are as follows:

1. Added experimental results for SDXL-Turbo and corrected a mistake in the BOPs calculation for SDv1.4 (Table 2).
2. Included evaluation results using IQA and a human preference reward model (Appendix C).
3. Added a discussion on Quantization-Aware Training (Appendix D.2).
4. Provided an explanation of Quantization Granularity (Appendix E).
5. Added qualitative results on the impact of the bit-width of the attention score quantizer (Appendix F).
6. Included statistical results to support the conclusion of Figure 5(b) (Appendix G).
7. Added full visualization of Figure 4(b) (Appendix H).
8. Provided qualitative results for the SDXL experiments (Appendix I).
9. Made minor revisions, including correction of typos.

We are currently conducting experiments on additional aspects that we were not able to include yet and plan to integrate them into the final version. Thank you.

---

### Author Response · Authors · 2024-12-04
**Summary of Rebuttal**

Dear Reviewers and AC,

We sincerely appreciate your valuable time and effort in reviewing our paper.

As the rebuttal period is coming to an end, we summarize the major concerns raised during the review and outline how they were addressed, respectively.

---

**Experiments on an advanced model:**

 - Reviewers **83s4**, **Hfso**, and **1KQn** recommended conducting experiments on advanced diffusion models. In response, we conducted additional experiments on SDXL-turbo. As a result, **our method significantly outperformed the existing baseline on SDXL-turbo**.



**Comparison with other methods(MixDQ, BitFusion):**

- Reviewers **Hfso** and **1KQn** asked about comparisons with MixDQ[1] and BitFusion[2], respectively. However, since these methods are based on mixed-precision (MixDQ) and quantization-aware training (BitFusion), **we believe they are not suitable for a direct apples-to-apples comparison with our fixed-precision, post-training quantization approach.** We compared our method with a strong baseline (TFMQ-DM[3]) under the same settings as our method. The results show that our approach achieves significantly higher performance.



**Additional Overheads:**

- There were comments about computational overheads incurred during $D_d$ operations (**s9vr**) and when processing the `<start>` token in full precision (**1KQn**). We clarified that **the $D_d$ operation is not performed during inference and thus it introduces no overhead**, and that the **overhead from processing the `<start>` token is only 0.0237 TBOPs, which is negligible**. Additionally, we are currently working on implementing the custom kernel ( as **qNA8**'s request ) and will add a comparison of actual inference performance to the final version. Furthermore, we will make our code publicly available.



**Evaluation with Additional Metrics:**

- Reviewer **1KQn** recommended evaluating image quality using IQA methods. In response, we evaluated the image quality using IQA methods (MANIQA[4]) and a human preference reward model (ImageReward[5]). The results show that **our method outperforms the existing baseline in these metrics as well**.



**Details of Figures:**

- Reviewers **Hfso** and **s9vr** requested the further information about some Figures (Figure 3, Figure 4(b), Figure 5(b)). In response, we have added detailed explanations in the appendix. Specifically, to better explain Figure 3, we included additional information about quantization granularity. For Figure 4(b), we included entire figure to the appendix. For Figure 5(b), R4 requested statistical results to support the conclusion. We have conducted additional experiments, confirming that the distribution of cross-attention indeed changes more dynamically.

---

During the rebuttal phase, we addressed most of the weaknesses and questions raised by the reviewers. As a result, reviewer **1KQn** raised their score from 5 to 8. Reviewers **qNA8** and **83s4** maintained their scores (6 and 8), both indicating a acceptance. Although two reviewers(**Hfso** and **s9vr**) did not respond, we would like to emphasize that we have clearly resolved the issues they raised.



We would like to highlight once again the contribution of our work. All reviewer recognized **our analysis and experiments** (**qNA8, 83s4, Hfso, s9vr, 1KQn**), and evaluated that **our proposed method and ideas are reasonable, clear, promising, innovative, or effective**(**83s4, HFso, s9vr**). Furthermore, it's **the first work to successfully achieve low-bit quantization of text-to-image diffusion models without requiring additional fine-tuning**.

---

We would like to express our sincere gratitude to all reviewers and AC for their efforts and time in reviewing our paper.

Best regards,
The Authors

---

[1] Zhao, Tianchen, et al. "MixDQ: Memory-Efficient Few-Step Text-to-Image Diffusion Models with Metric-Decoupled Mixed Precision Quantization." *arXiv preprint arXiv:2405.17873* (2024).

[2] Sui, Yang, et al. "BitsFusion: 1.99 bits Weight Quantization of Diffusion Model." *arXiv preprint arXiv:2406.04333* (2024).

[3] Huang, Yushi, et al. "Tfmq-dm: Temporal feature maintenance quantization for diffusion models." *Proceedings of the IEEE/CVF Conference on Computer Vision and Pattern Recognition*. 2024.

[4] Yang, Sidi, et al. "Maniqa: Multi-dimension attention network for no-reference image quality assessment." *Proceedings of the IEEE/CVF Conference on Computer Vision and Pattern Recognition*. 2022.

[5] Xu, Jiazheng, et al. "Imagereward: Learning and evaluating human preferences for text-to-image generation." *Advances in Neural Information Processing Systems* 36 (2024).

---

### Meta-Review · Area_Chair_Zsqa · 2024-12-16

**Metareview:**

This paper discovered that the activation outlier and cross-attention scores plays an important role in image quality and alignment with texts in T2I models, and then proposed an effective approach (DGQ) to solve the quality degradation issue when quantizing diffusion models. All reviewers are consistently positive about this work (with two clear accept and three borderline accept) by appreciating the solid analysis and effectiveness of solution. Most concerns focus on a few aspects of evaluation and clarity of figures, which are addressed via detailed rebuttal from authors. While some reviewers did not respond, the AC helped go over the rebuttal and agreed that the proposed concern is mostly addressed. Overall, this is a good work by solving a known issue in model quantization with a simple yet effective training-free approach and of great value to community to push forward this direction. Considering all reviews and discussions, a decision of acceptance is made and authors are suggested to include all the suggestions in the final revised version.

**Additional Comments On Reviewer Discussion:**

The main points raised by reviewers lie in evaluation including comparisons, building stronger pipeline and more metrics. Through rebuttal and discussion, authors provided detailed feedback to address those concerns. As a result, reviewer 1KQn raised their score from 5 to 8. Reviewers qNA8 and 83s4 maintained their scores (6 and 8), both indicating a acceptance. Although two reviewers (Hfso and s9vr) did not respond, AC went over the reply from authors and agreed that it is informative and clarifies the concerns as well. Overall the reviews are unanimously positive and the work looks more complete with authors' respond, and thus AC made the accept decision.

---

### Decision · Program_Chairs · 2025-01-22

Accept (Poster)